# Thunderstorm Ground Enhancements Measured on Aragats and Progress of High-Energy Physics in the Atmosphere

Ashot Chilingarian 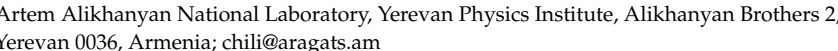

Artem Alikhanyan National Laboratory, Yerevan Physics Institute, Alikhanyan Brothers 2, Yerevan 0036, Armenia; chili@aragats.am

**Abstract:** High-energy physics in the atmosphere (HEPA) has undergone an intense reformation in the last decade. Correlated measurements of particle fluxes modulated by strong atmospheric electric fields, simultaneous measurements of the disturbances of the near-surface electric fields and lightning location, and registration of various meteorological parameters on the Earth have led to a better understanding of the complex processes in the terrestrial atmosphere. The cooperation of cosmic rays and atmospheric physics has led to the development of models for the origin of particle bursts recorded on the Earth's surface, estimation of vertical and horizontal profiles of electric fields in the lower atmosphere, recovery of electron and gamma ray energy spectra, the muon deceleration effect, etc. The main goal of this review is to demonstrate how the measurements performed at the Aragats cosmic ray observatory led to new results in atmospheric physics. We monitored particle fluxes around the clock using synchronized networks of advanced sensors that recorded and stored multidimensional data in databases with open, fast, and reliable access. Visualization and statistical analysis of particle data from hundreds of measurement channels disclosed the structure and strength of the atmospheric electric fields and explained observed particle bursts. Consequent solving of direct and inverse problems of cosmic rays revealed the modulation effects that the atmospheric electric field has on cosmic ray fluxes.

**Keywords:** atmospheric electric field; relativistic runaway electron avalanche; thunderstorm ground enhancement; extensive air shower; Forbush decrease

## 1. Introduction

Various particle accelerators operate in the cosmic plasma, filling the galaxy with high-energy particles (primary cosmic rays). Reaching the Earth's atmosphere, these particles cause extensive air showers (EASs) consisting of millions of elementary particles (secondary cosmic rays), covering several km$^2$ on the ground. During thunderstorms, strong electric fields modulate the energy spectra of secondary particles and cause short and long particle bursts. Large amplifications of particle fluxes (the so-called thunderstorm ground enhancements (TGEs, [1,2]) manifest themselves as prominent peaks in the time series of count rates of particle detectors, coinciding with a strong atmospheric electric field accelerating and multiplying the free electrons of cosmic rays. Free electrons, abundant at any altitude in the atmosphere from the small to large EASs, serve as seeds for atmospheric electron accelerators, an analog of "electron guns" in artificial accelerators. EAS cores randomly hitting arrays of particle detectors generate short bursts of relativistic particles with a duration of fewer than 1 µs [3].

To detect bursts of both types of particles, networks of particle detectors and spectrometers are required, which have operated reliably for many years [4–6]. Long-term observations of the Aragats cosmic ray observatory provide massive data on particle fluxes and bursts. Comparing the energy spectra of electrons and gamma rays measured at Aragats makes it possible to identify emerging electrical structures in the atmosphere, which accelerate seed electrons to 50 MeV or more [7,8]. Massive fluxes of electrons and

gamma rays, measured on mountain peaks, can exceed the background up to 100 times. The long-term impact of this radiation needs to be carefully assessed.

In turn, if the purpose of the experiment is to accurately measure the size of the EAS (the number of electrons), it is necessary to take into account the influence of atmospheric electric fields that increase the number of shower particles and introduce a bias in the estimate of the energy of the primary particle. The existence of such a strong electric field above EAS arrays is proved by detecting an abrupt increase in the number of EASs during thunderstorms that meet the trigger conditions (up to 20%) reported by surface arrays [9–11]. Modeling confirms that free electrons from EASs, entering strong atmospheric electric fields, generate multiple electron–photon avalanches that induce additional surface array triggers [12]. Muons do not multiply in an electric field like electrons; however, electric fields lead to the modernization of their energy spectra and produce an interesting muon-stopping effect, which is used to estimate the maximum energy of atmospheric electric fields [13].

At Aragats, special attention was paid to the joint monitoring of particle fluxes and atmospheric discharges. Multiyear monitoring results and modeling of electron–gamma avalanches made it possible to formulate hypotheses on the origin of particle bursts. Until now, two main models have been proposed to explain particle bursts: a relativistic avalanche of runaway electrons (the RREA model) [14–16] and the model of particle generation in or around a lightning discharge. However, even though "lightning discharges are currently recognized as powerful particle accelerators" (see [17,18] and references therein), the physical mechanism of how the discharge produces a huge number of relativistic particles is still under discussion (see proposed scenarios in [19,20]). To resolve this contradiction, it is necessary to unequivocally answer the question: do lightning flashes emit electrons, positrons, gamma quanta, and neutrons with energies of several tens of MeV? Recently, research teams using large arrays of particle detectors deployed to detect EASs have become interested in unusual triggers (bursts of particles called downward terrestrial gamma-ray flashes (DTGFs)) and their possible correlations with thunderstorm activity. The origin of these controversial events can be clarified by invoking well-known knowledge on the registration of the EAS cores.

Violent solar bursts fill the interplanetary space with immense magnetized plasma structures, moving up to 3000 km/s (the so-called interplanetary coronal mass ejection (ICME)) and perturbing the interplanetary magnetic field (IMF) and the magnetosphere. These disturbances could lead to major geomagnetic storms damaging multi-billion-dollar assets in space and on Earth. Monitoring the high-energy particles can provide highly cost-effective information also for predicting geomagnetic storms.

For fundamental research in solar physics, solar–terrestrial relations, and space weather, as well as for forecasting the dangerous consequences of space storms, networks of particle detectors located in different geographical coordinates and measuring various types of secondary cosmic rays are of vital importance. Geophysical research is becoming increasingly important in the coming decades when natural disasters are rising. Solar, astrophysical, and atmospheric physics are synergistically linked and need to be integrated to reveal consequences of violent solar flares, and extreme atmospheric electric fields. The synergy of high-energy space and atmospheric physics will open up new research areas for a better understanding and development of geospace physics.

## 2. Model of a Thunderstorm Ground Enhancement (TGE)

Various types of cosmic rays, including electrons, muons, gamma rays, and neutrons, are new messengers detailing information about the charge structure of the thunderous atmosphere. Atmospheric electric fields and atmospheric discharges have been intensively studied in recent decades using radars, 3D lightning map arrays (LMA, [21]), worldwide lightning location networks (e.g., WWLLN, [22]), and very-high-frequency interferometer systems [23]. These measurements are coupled with synchronous measurements of per-

turbations of the near-surface electric field (NSEF) by a network of sensors (for example, EFM-100 electric field sensors from BOLTEK company) [24].

Experiments conducted between 1945 and 1949 at the Zugspitze observatory in Germany [25] revealed a rather complex structure of the intracloud electric field. Joachim Küttner discovered a pocket of positive charge (lower positive charge region (LPCR)) at the base of the cloud and coined the term "Graupel dipole", a charge structure formed by the LPCR and the main negatively charged layer. A thundercloud's charged-layer localization can be quite complex (see [26], for instance). However, a simplified structure with two main charged regions (positive over negative) and a relatively small LPCR is usually accepted as a basic tripole structure. Tsuchiya [27] suggested that short-lived tripole structures appeared in the thundercloud during winter thunderstorms in Japan. In recent years, several tens of TGEs have been registered in Japan (the authors call them gamma glows [28]).

A model of the charge structure of a thundercloud is shown in Figure 1. The left side shows electron avalanches that develop in the lower dipole (TGEs) and the upper dipole of a thundercloud (the so-called terrestrial gamma-ray flashes, TGF, [29]). The RREA is a threshold process that occurs only if the atmospheric electric field exceeds the critical value in a region of a vertical extent of 1–2 km. The threshold (critical) energy smoothly increases with the increase in air density according to $E_{th} = n \times 2.84$ kV/cm [15,16], where n is the relative air density, i.e., the air density normalized to the sea level value, that equals 1.225 kg/m$^3$. The following scenarios of electron acceleration in the atmospheric electric fields can be considered:

1. The dipole formed by the main negative layer in the middle of the thundercloud (MN) and its mirror image (hereafter, MN-MIRR) accelerates electrons downward. The electric field can extend close to the Earth's surface, with gamma rays and electrons registered by particle detectors. The NSEF falls in the negative domain, reaching $\approx -30$ kV/m for the largest TGEs.

   LPCR is a transient structure sitting on the graupel pallets, disappearing with graupel fall.

2. In addition to the MN-MIRR, another dipole is formed by MN-LPCR. For a few minutes, when LPCR is large enough, it screens the detector site from the negative charge of MN, and a positive NSEF is observed.

TGEs are very intense in spring and autumn when the LPCR can be very close to the Earth's surface ($\approx 50$ m). Fields induced by the MN-mirror and MN-LPCR are identically directed, and their sum can reach relatively large values exceeding the threshold value to start RREA by 20–30%. In summer, the distance to the cloud base is more prominent (150–400 m); usually, only gamma rays reach the Earth's surface and are registered.

In addition to these basic scenarios, the cloud's fast-changing charge structure produces more complicated electric field configurations. For instance, TGE can start with mature LPCR; the near-surface electric field reaches positive values for a few minutes and returns to deep negative values when LPCR shrinks.

Lightning flashes reduce the negative charge above the Earth's surface, reducing the electric field strength below the RREA initiation threshold. Consequently, the TGE abruptly stops. However, a smaller near-surface electric field still exists, and $^{222}$Rn offspring amplify the "background" gamma-ray flux, initiating long-term TGEs [30]. Moreover, the TGE continues after the return of the near-surface electric field strength to the fair-weather value due to tens of minutes of the lifetime of the $^{214}$Pb and $^{214}$Bi isotopes. Rain returns the $^{222}$Rn progeny from the atmosphere to the Earth's surface and provides additional gamma radiation for several tens of minutes (Radon circulation effect, see [31]).

From 2009 to 2022, about 600 TGEs were registered at the Aragats (see TGE catalogs in [32,33]). Numerous particle detectors and field meters operating year-round provide continuous recording of the time series of charged and neutral particle fluxes. Accurate measurements of TGE fluxes are made using a large scintillation spectrometer capable of recovering the energy spectra of electrons and gamma rays separately. By comparing

the simulated and reconstructed energy spectra, we show that the modeled values of the electric field strength afford particle fluxes entirely consistent with the measured ones, thereby proving the correctness of the origin of the TGE phenomenon from RREA. The recovered spectra of TGE particles show the modulation effect of the atmospheric electric field on the cosmic ray population, disclosing its strength and location. To confirm the charged structures of the thundercloud, we compare estimates of the atmospheric electric field, obtained using modulated particle fluxes, with the density distribution of hydrometeors in a thundercloud, obtained from the Weather Research and Forecasting Model (WRF, https://www.mmm.ucar.edu/weather-research-and-forecasting-model, accessed on 29 January 2023). Another possibility to estimate the density distribution of hydrometeors is to use the S-band radar, located ≈20 km from Aragats station [34].

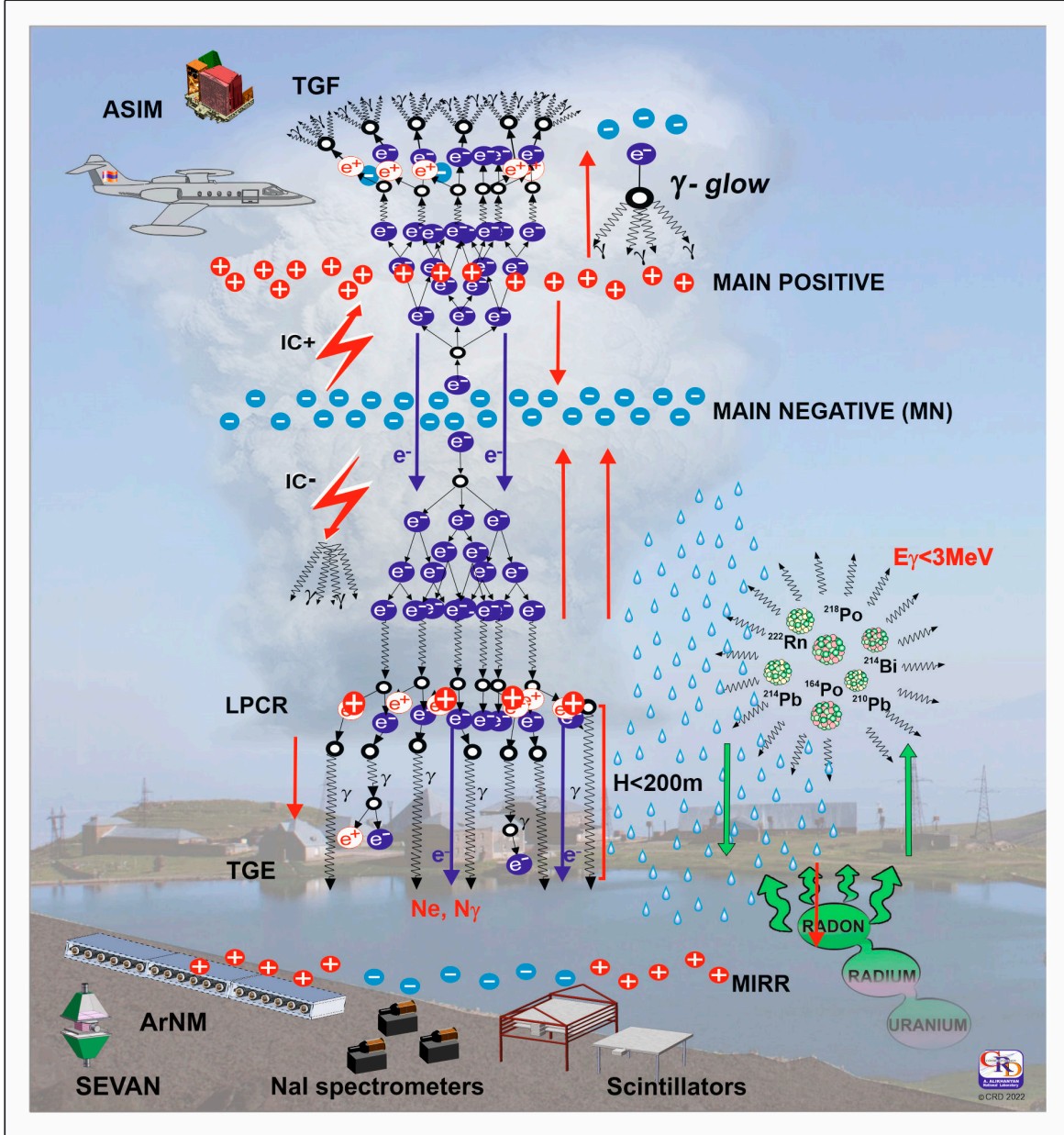

**Figure 1.** Charged structures and particle fluxes associated with a thunderstorm. On the left are particle avalanches developing in a thundercloud, and on the right are emissions of $^{222}$Rn descendants. The red arrows show atmospheric electric fields and lightning flashes. ©PRD. Reproduced with permission.

### 3. Electron Energy Spectra

The ultimate proof that the origin of TGE is an RREA developing in the thunderous atmosphere above detectors is recovering the electron energy spectrum. However, it is a rather difficult task; after leaving the region of a strong electric field, where the electrons are accelerated and multiplied, the intensity of the electron beam decreases rapidly due to ionization losses. By contrast, gamma rays are attenuated much slower; thus, at the Earth's surface, the intensity of TGE gamma rays usually is much higher than that of electrons. Available NaI spectrometers have a relatively small area (0.01–0.03 m$^2$) and limited energy range (below 10 Mev); thus, they measure mostly the flux of gamma rays with a negligibly small fraction of electrons. Therefore, the only spectrometer observing the fluxes of RREA electrons and recovering their energy is the Aragats solar neutron telescope (ASNT, Figure 2); see [4] for details.

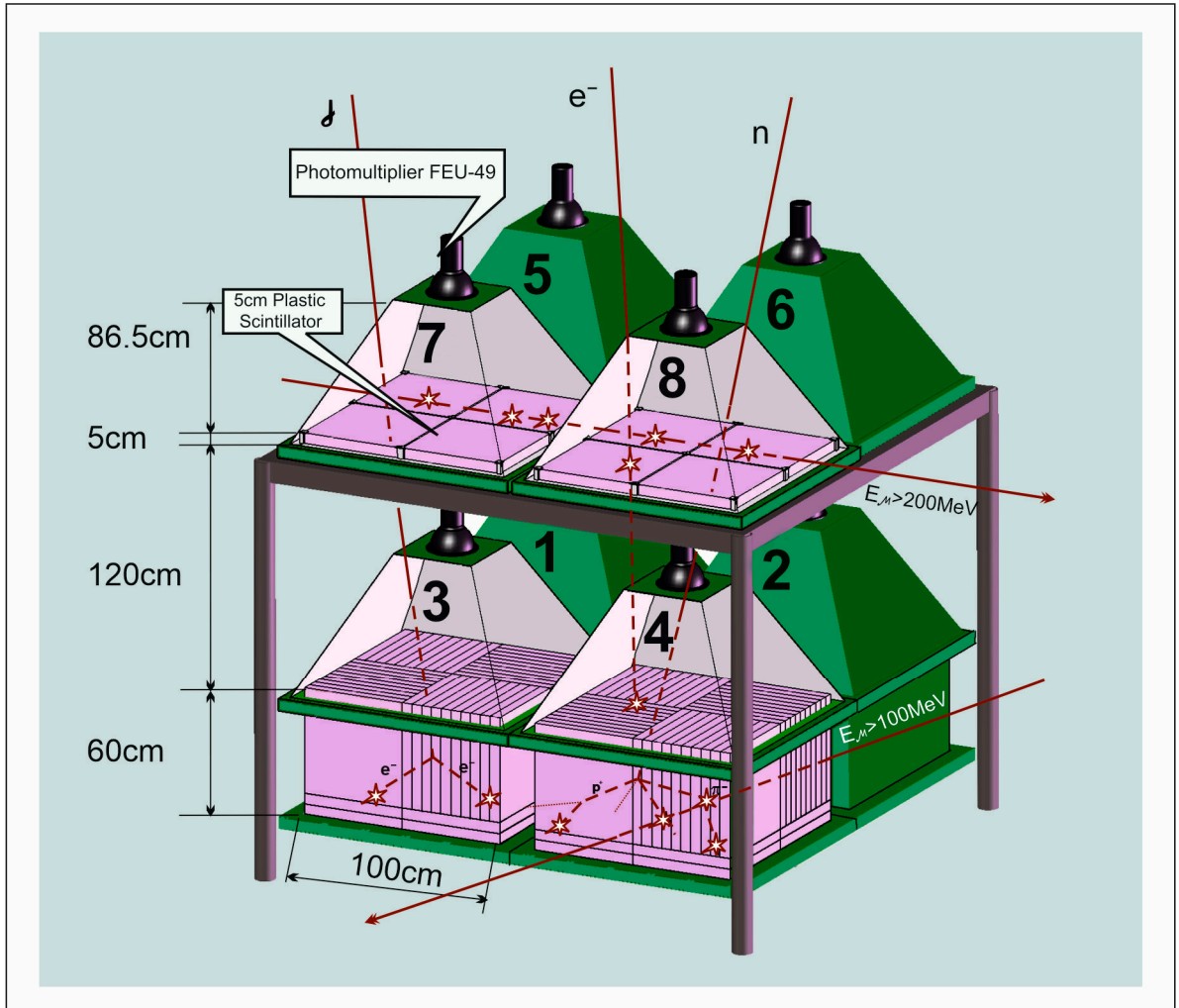

**Figure 2.** Assembly of ASNT with a schematic view of the different particles traversing the detector. Both upper and lower scintillators register TGE electrons; gamma rays and neutrons are registered by invoking the veto option (no signals in the upper scintillators). ©PRD. Reproduced with permission.

Only in 2018, after numerous simulations of particle transport through the atmosphere and ASNT detector with GEANT4 package [35], and after developing a technique for solving the inverse problem of cosmic rays, we confirmed the possibility of obtaining the energy spectra of TGE electrons and gamma rays separately. The ASNT spectrometer consists of a 60 cm thick plastic scintillator with a 4 m$^2$ area (100 times larger than the largest NaI crystal used in HEPA measurements) and a 5 cm thick "veto" scintillator with

the same area. From the ASNT detector, we obtained a 2 s time series of count rates of each of 8 scintillators and a 20 s time series of energy release histograms in 5 and 60 cm thick scintillators.

Coincidences of signals from four upper and four lower scintillators are counted separately, thus roughly estimating the electron incidence angles. Figure 3 shows the count rates of particles arriving in the 0–22° cone (black curve) and the 22–58° zenith angles from 4 different azimuth angles of 0, 90, 180, and 270 degrees (colored curves). As we see in Figure 3, the TGE particles arrive vertically as a vertical electric field accelerates the electrons.

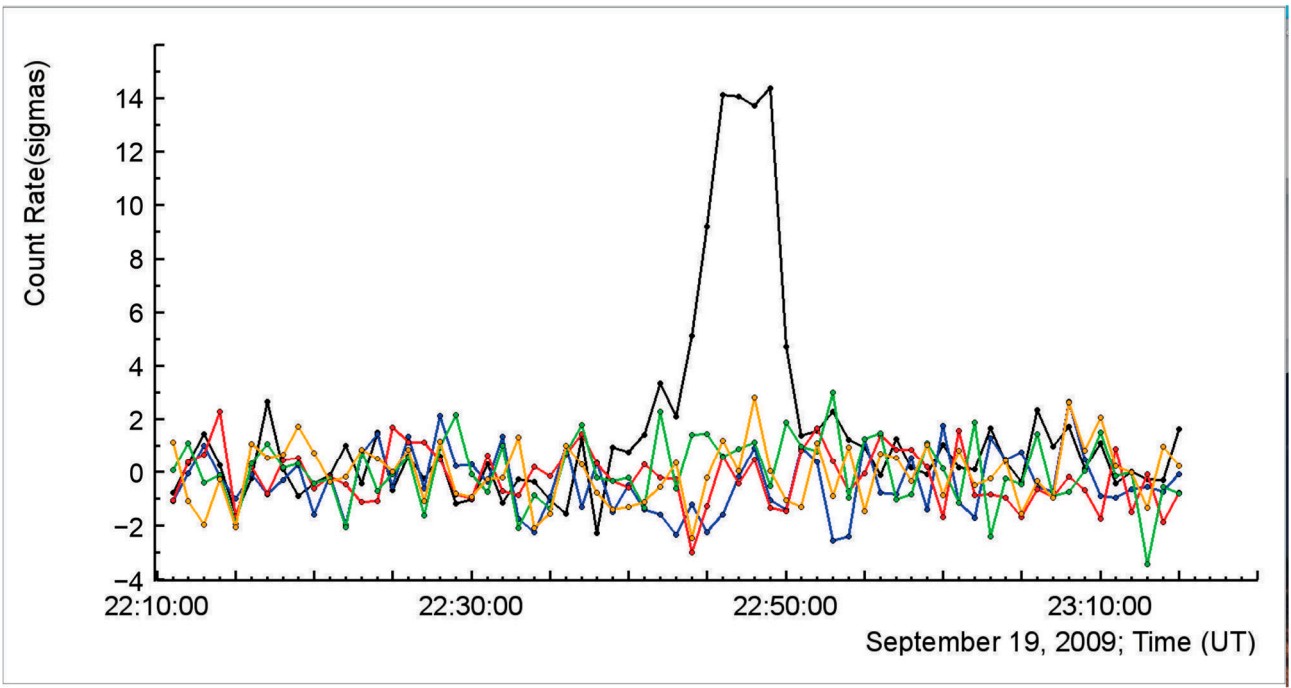

**Figure 3.** Time series of the count rates of particles arriving from different directions; black—vertical direction (coincidences of scintillators 1-5, 2-6, 3-7, and 4-8); blue—direction connected 1-6 and 3-8 scintillators; red—direction 1-7 and 2-8; green—direction 2-5 and 4-7; and yellow—direction 3-5 and 4-6. ©PRD. Reproduced with permission.

Figure 4a,c shows the time series of the coincidence of "01" (signal only in the lower layer) and "11" (signals also in the upper layer) for the 1 min and 2 s count rates (in 2009, the ASNT electronics sampling time was 1 min). To demonstrate the significance of peaks, we show the time series in units of standard deviations relative to the mean value measured in fairweather. The electron detection efficiency of a 5 cm scintillator is ≈95% and 5–7% for gamma rays; for the 60 cm scintillator, the detection efficiency is above 95% for electrons and 40–60% for gamma rays. Thus, particles registered by "01" coincidence are mostly gamma rays, and by "11" coincidence, they are primarily electrons. The large $N_e/N\gamma$ ratio shown in Figure 4a,c at minutes 2:03–2:04 and 22:49–22:50 allows the electron energy spectra recovery. Figure 4b,d shows the restored differential energy spectra of electrons and gamma rays (see [4] for details of the spectrum recovering method). The intensity of the electron flux measured on 6 October 2021 is significantly lower than the intensity of gamma rays (Figure 4b).

On the contrary, as seen in Figure 4d, the intensities of the electron and gamma rays are almost equal for the TGE measured on 19 September 2009. The ratio of electrons to gamma rays reaches 66% at minutes 22:49–22:50. This can only occur if the strong accelerating electric field is very low above the ground (we estimate a height of 25–50 m). The electron avalanche covers a significant area on the ground. Additional evidence of electron flux registration on 19 September 2009 is shown in Figure 5.

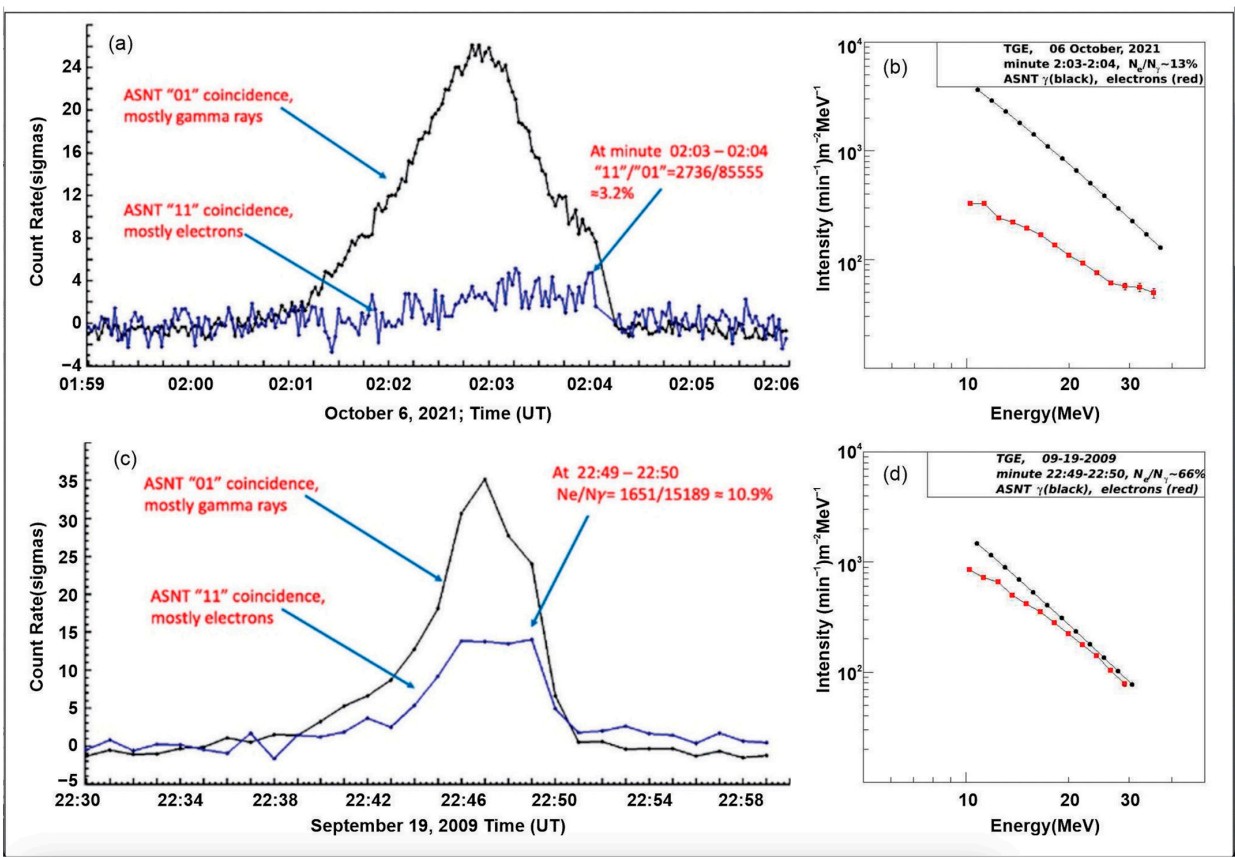

**Figure 4.** (**a**) 2 s time series of the count rate coincidences measured using the ASNT spectrometer, shown in the number of standard deviations (Nσ) from the mean value measured in fairweather before the TGE; (**b**) differential energy spectra recovered from the energy release histograms; (**c**) 1 min time series of the coincidence count rates measured using the ASNT spectrometer; (**d**) differential energy spectra recovered from energy release histograms. ©PRD. Reproduced with permission.

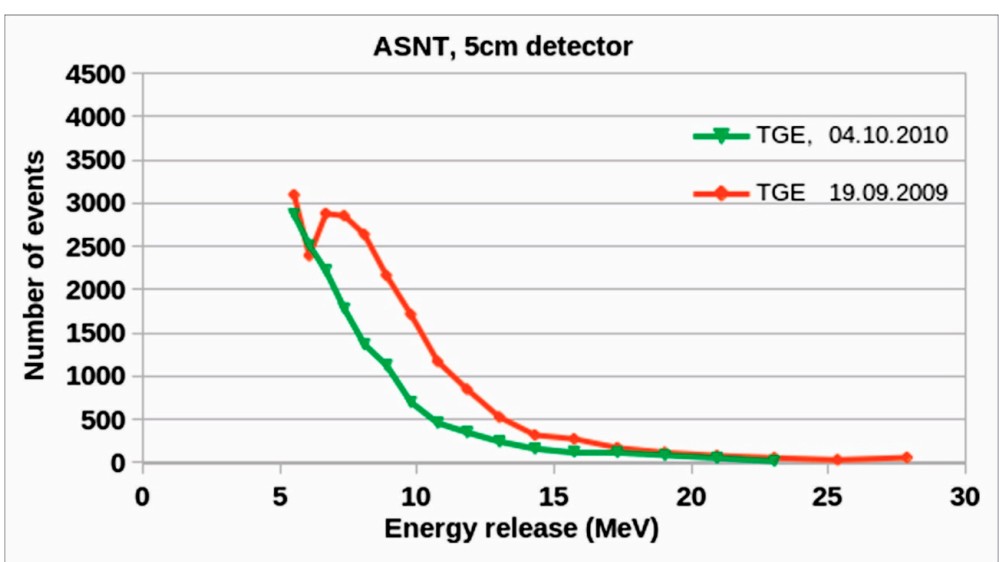

**Figure 5.** Energy release histograms in the 5 cm thick scintillator were measured on 19 September 2009 and 4 October 2010. ©PRD. Reproduced with permission.

Figure 5 shows histograms of energy release in the 5 cm thick scintillator at minutes 2:49–22:50 on 19 September 2009 (red curve), which peaked at 6–8 MeV because the electron

ionization losses in the scintillator are ≈1.8 MeV per centimeter. Thus, only high-energy electrons (with energies of 10 MeV or more) can cause this peak in the energy-release histogram. The energy release histogram caused by TGE electrons is smeared by the energy release of gamma rays (gamma ray detection efficiency is 5–7%). However, the gamma ray energy release has an exponential form. It does not contribute to the histogram's peak, as shown in Figure 5 (green curve). The TGE observed on 4 October 2010 contains no electrons because the electric field terminates high above the Earth's surface, and the electron flux did not reach the spectrometer. Thus, if we have a prominent peak in the time series of coincidence "11" (Figure 4a,b) and a peak in the energy release histogram (red curve in Figure 5), we can be sure that the TGE contains a significant fraction of electrons. And it will be possible to recover the energy spectrum as was performed for the 19 September 2009 and 4 October 2021 TGEs (red curves in Figure 3b,d).

## 4. Vertical and Horizontal Profiles of the Atmospheric Electric Field during Thunderstorms

To understand the avalanche development in the electrified atmosphere and to compare measured energy spectra with simulated ones, we used the CORSIKA code [36], which considers the electric field's effect on the transport of particles. Figure 6 shows the simulation of the RREA development at different depths in the atmosphere and for different physically grounded intracloud electric field strengths (see [37] for details). The CORSIKA code tracks RREA particles and calculates the number of electrons and gamma rays at different stages of avalanche development in a strong electric field and an additional 200 m. The curves are normalized to one m² and one seed electron for easier comparison with integral spectra recovered from the energy release histograms registered by the ASNT detector. The numerical data corresponding to Figure 6 is placed in Table 1. For low electric field intensities (1.8 and 1.9 kV/cm), the RREA process attenuates before reaching the observation level at 3200 m (see green and brown curves in Figure 6a) because the electric field critical value at 4000 m is ≈1.9 kV/cm and RREA process below 4000 m stops.

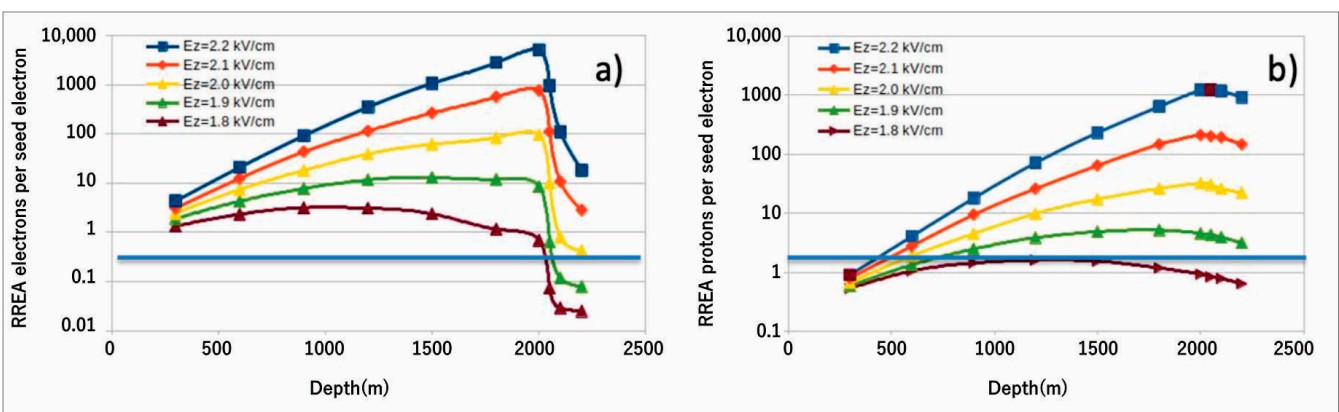

**Figure 6.** Development of the RREA in the atmosphere (**a**) electrons (**b**) gamma rays. The avalanche began at an altitude of 5400 m above sea level (depth 0), i.e., 2200 m above the Aragats station. The number of avalanche particles is calculated every 300 m. After leaving the electric field, the tracking of avalanche particles continues another 200 m until reaching the ground. The blue line shows the number of electrons and gamma rays per seed electron per m² measured on 14 June 2020 (see Table 1).

The propagation of electrons and gamma rays in the avalanche was tracked until their energy decreased to 0.05 MeV. The energy spectrum (from 1 to 300 MeV) and the number of the seed electrons were taken from the WEB calculator EXPACS [38] for the Aragats geographical coordinates and height of 5400 m. The simulations include 10,000 events for electric field strengths of 1.8–2.2 kV/cm.

By comparing the numbers of the recovered electrons and gamma rays (ASNT integral energy spectra) with simulations, we can select plausible parameters of the RREA that can

reproduce the TGEs of summer 2020; see Table 1 and Figure 6. In the first two columns of Table 1, we show the intracloud electric field parameters used in the simulations and the date of the TGEs, which were compared with simulations. In the third column, we show the electric field termination height. The number of electrons and gamma rays for different electric field strengths and termination heights, as well as numbers recovered from ASNT energy spectra, are shown in the fourth and fifth columns. The electron and gamma ray number per seed electron were obtained by integrating the recovered differential energy spectrum and by dividing the obtained value by the number of seed electrons with energies above 1 MeV (42,000 per $m^2$ per minute at 5400 m). From Table 1 and Figure 6, we see that the relatively small TGEs (14 June 2020) fits the simulations of RREAs developing in electric fields of 1.8–1.9 kV/cm, which end at $\approx$100 m above the ground. RREA developed in the field of intensity 2.0 kV/cm produces much more gamma rays per seed electron (22) than measured in TGE. A total of 2 smaller TGEs in the summer of 2020 can be related to the simulations with a 1.8 kV/cm electric field terminated above 100 m.

**Table 1.** Parameters of simulated RREAs, calculated with CORSIKA code, and of three TGEs observed in 2020 (last three rows).

| Atm. El. Field | Date of TGE | Height of El. Field Termination | N of El. E > 4 MeV per Seed Electron | N of $\gamma$ Rays E > 4 MeV per Seed Electron |
|---|---|---|---|---|
| 1.8 kV/cm | - | 100 | 0.03 | 0.78 |
| 1.9 kV/cm | - | 100 | 0.12 | 3.9 |
| 1.9 kV/cm | - | 200 | 0.08 | 3.1 |
| 2.0 kV/m | - | 200 | 0.43 | 22 |
| - | 14 June 2020 | - | 0.14 | 1.26 |
| - | 27 June 2020 | - | 0.041 | 0.51 |
| - | 23 July 2020 | - | 0.059 | 0.49 |

Of course, we cannot expect the exact coincidence of the simulated and measured count rates because of the specific atmospheric conditions for each measured TGE and several oversimplified assumptions in the simulation. Nevertheless, we can conclude that the modeled RREA process within the chosen electric field strengths can explain the TGE phenomena.

TGE monitoring on Mount Aragats also provides clues for estimating the horizontal expansion of the atmospheric electric field during thunderstorms. Using the STAND1 particle detector network [5], we measured 1 s time series of count rates on an area of $\approx 10^5$ $m^2$. The scatter plots show that the NSEF was uniform and stable during the TGE on 22 May 2018; see Figure 7. The correlation coefficients between the flux enhancements measured by the particle detectors are pretty significant (Figure 7a,c). As seen from Figure 7b,d (follow the red arrows), there is no delay in the correlation plots (the top of the delay curve is flat and constant within several seconds; thus, we can accept that the time lag is near zero).

To obtain an idea of the atmospheric electric field over longer distances, we installed networks of SEVAN detectors, NaI spectrometers, and electric field sensors on Mount Aragats [39]. On the first of May 2022, the storm began in Nor Amberd around noon UT and on Aragats half an hour later (the start of the NSEF disturbances) and ended at $\approx$13:30 and 14:00 UT, respectively. Figure 8a,b shows a 1 min time series of count rates measured with 5 cm thick and 1 $m^2$ area plastic scintillators (SEVAN detector's upper scintillators) located under a 0.8 mm thick iron roof. TGEs occurred at Aragats at 13:00–13:14 UT and 13:23–13:33 UT and at Nor Amberd at 12:30–13:23 UT. The values of the highest peaks were 5.7% (9$\sigma$) and 6.6% (8.2$\sigma$) at Aragats and Nor Amberd, respectively. Thus, the atmospheric electric field in the clouds above both stations (distance of 13 km) exceeded the critical value for activating the runaway process. The electron accelerator worked for several minutes, sending particle avalanches toward the Earth's surface. The NSEF varies during the TGE from $-23$ to 8 kV/m at Aragats and from $-25$ to 25 kV/m at Nor Amberd. The TGE occurred on Aragats during the deep negative electric field and Nor Amberd during

the positive NSEF. Thus, despite different conditions of the NSEF perturbations and other charge structures in the cloud above it, the maximum particle fluxes measured at both stations with the same type of detectors occurred within ≈20 min.

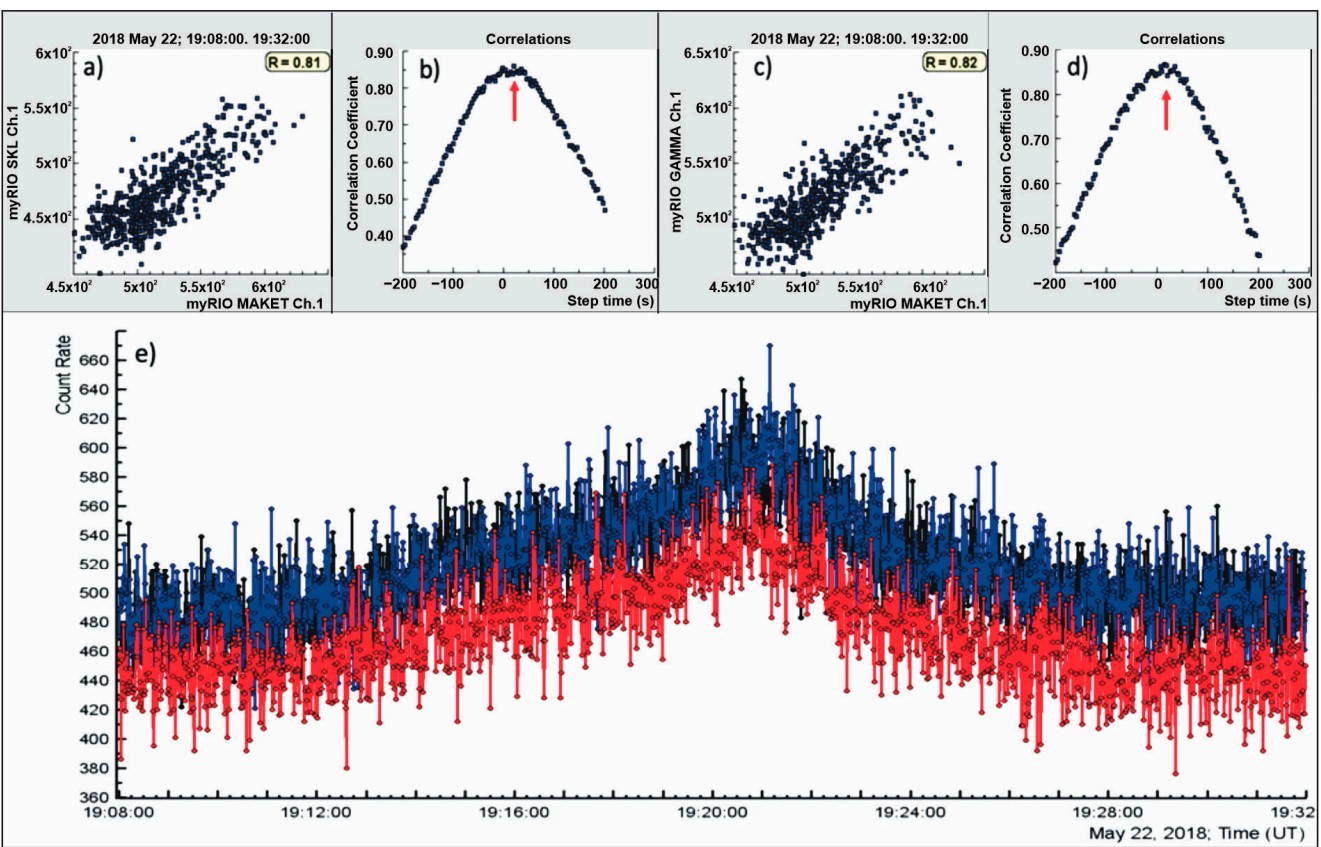

**Figure 7.** (**a**,**b**) scattering plot and plot of lagged correlations of count rates of upper scintillators of the 1 cm thick STAND1 detector (MAKET and SKL detectors); (**c**,**d**) the same for MAKET and GAMMA detectors; (**e**) 1-second time series of all three units of STAND1 network: MAKET (black), GAMMA (blue) and SKL (red).

Another strong thunderstorm occurred at Aragats on 6 September. The character of NSEF disturbances was approximately the same at both stations; see Figure 9a. The strengthening of the particle flux began at both stations simultaneously at 15:59 UT; see Figure 9b. The maximum of counts at the two identical particle detectors was reached at Nor Amberd at 16:01 UT and Aragats at 16:07 UT and 16:08 UT. Large peak values for all 4 sensors leave no doubt that TGE occurred at both stations, located at a horizontal distance of 13 km and an altitude difference of 1.2 km, almost simultaneously.

In Figure 10 we present an episode from a very long storm that occurred on 22 September 2022. This time the pattern of NSEF disturbances was different on both stations. On Aragats again multiple lightning flashes were observed, and in Nor Amberd there were a few flashes but the amplitude of NSEF fluctuations was rather large, see Figure 10a. The TGE was measured with NaI(Tl) detectors. At this time a large flux was observed also in Burakan, 3 km far and 200 m lower than Nor Amberd. Close occurrences of flux enhancements in time and very large significances of peaks leave no doubt that this time avalanches covered an even larger area than on 6 September extended by 15 km.

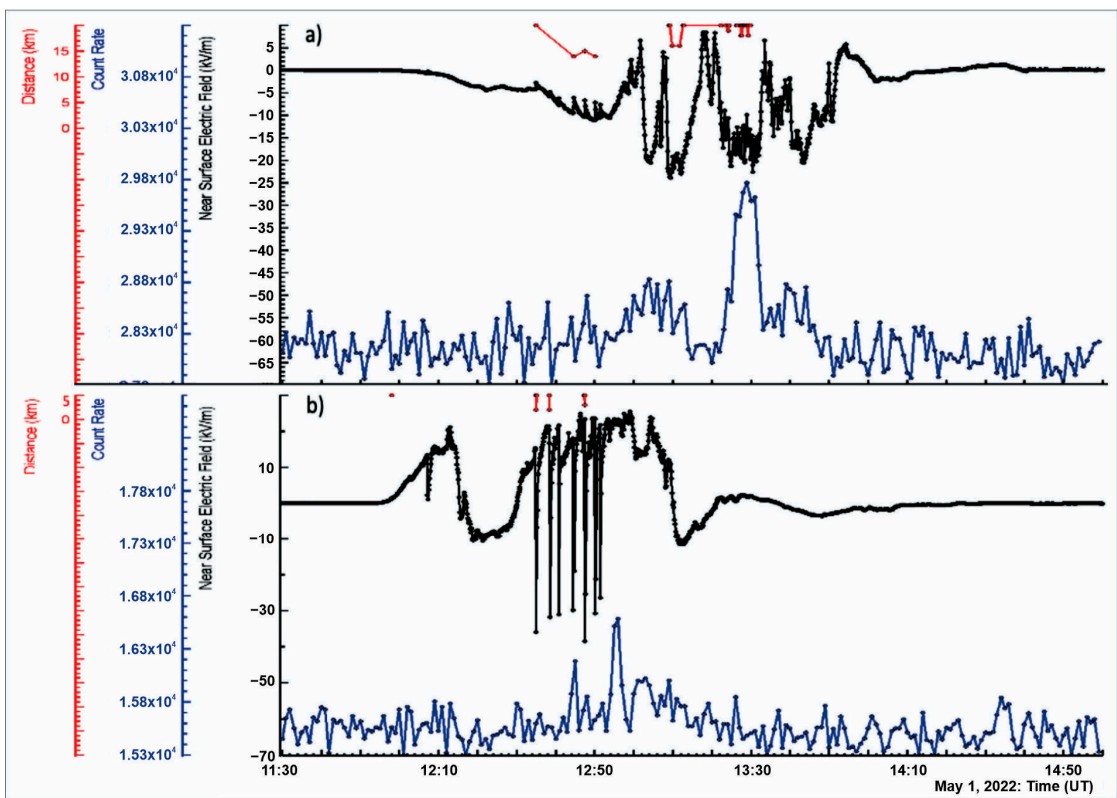

**Figure 8.** Disturbances of the NSEF (black); 1-minute time series of count rates of 5 cm thick and 1 m$^2$ area plastic scintillators (blue); and distances to lightning flashes (red) measured on Aragats (**a**) and in Nor Amberd (**b**).

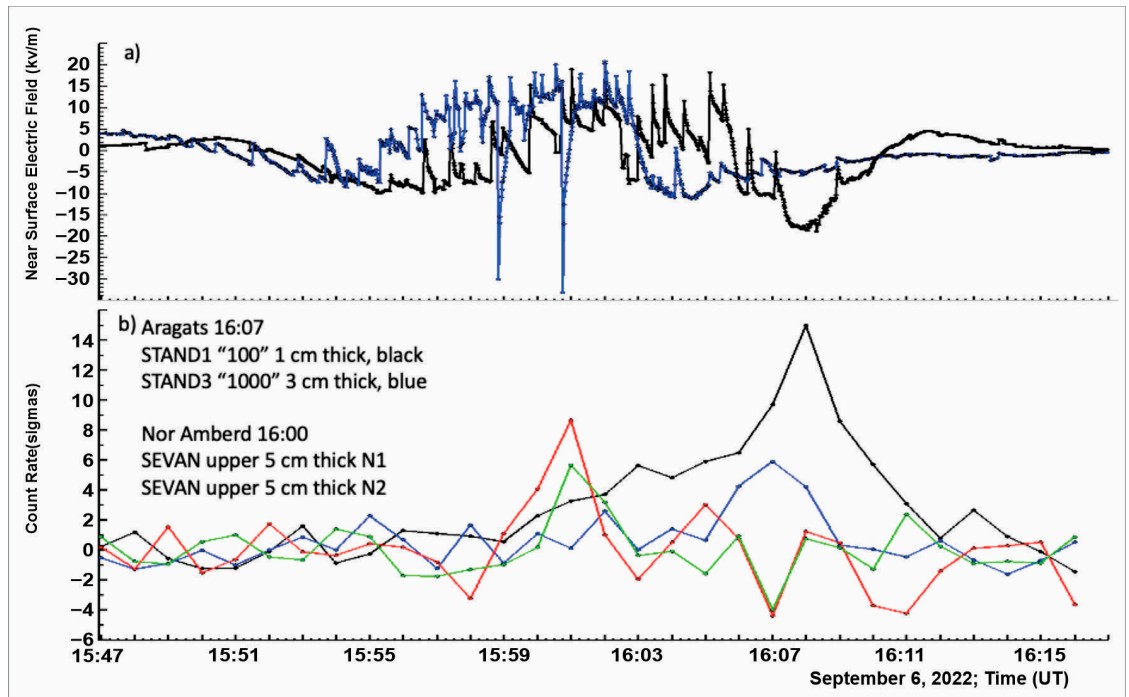

**Figure 9.** (**a**) Disturbances of the NSEF measured on Aragats (black) and in Nor Amberd (blue); (**b**) 1-min time series of count rates of 1 cm and 3 cm thick (both 1 m$^2$ area) plastic scintillators on Aragats (black and blue) and 2 identical 5 cm thick and 1 m$^2$ area plastic scintillators in Nor Amberd (red and green).

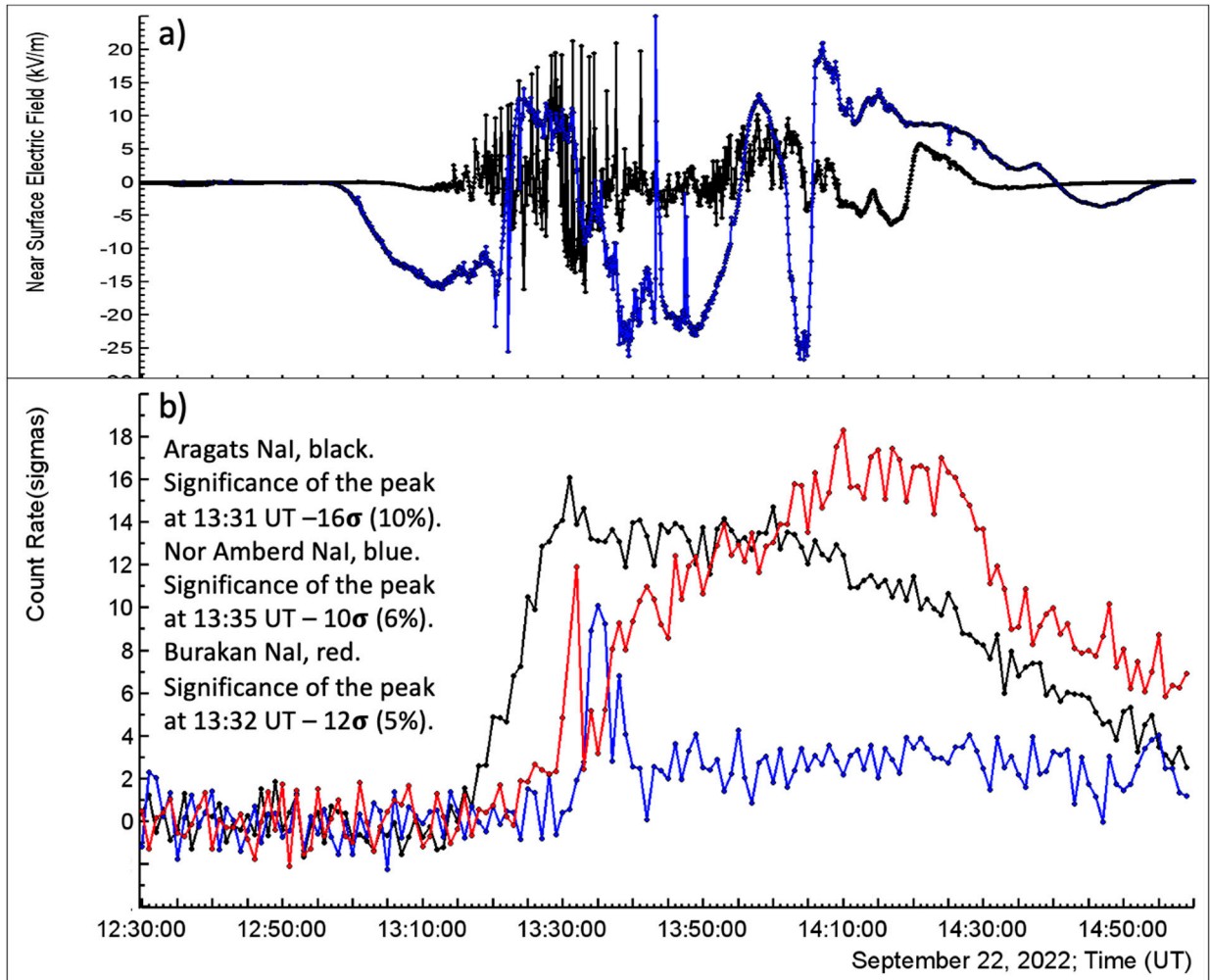

**Figure 10.** (**a**) NSEF perturbations measured at Aragats (black) and Nor Amberd (blue); (**b**) 1-min time series of count rates of plastic scintillators 5 cm thick and 1 m$^2$ in area, on Aragats (black), in Nor Amberd (blue) and in Burakan (red).

Figure 11 shows additional evidence for the occurrence of the TGE in Nor Amberd. To record the spatial expansion of an individual RRE avalanche (generated by a single seed electron), we measure signal co-incidences of 3 detectors located over an area of 3 m$^2$ (within a 1-microsecond window). During 6 min of TGE (from 13:34 UT to 13:40 UT), the count rate of this coincidence increased dramatically from ≈20 to 62 (at 13:36 UT), see Figure 11. Thus, the electron-gamma avalanches covered an area of 3 m$^2$. We have already shown that the RREAs origi-nated in a thunderous atmosphere (from multiple seed electrons) coming one after another cover an area as large as several square kilometers. Here we register TGEs from a single seed electron, which gives rise to avalanche particles arriving within one microsecond. On 19 September 2009, the size of TGE from one single electron (particles registered within 1 micro-second) was estimated by the MAKET array to be several hundred square meters [2].

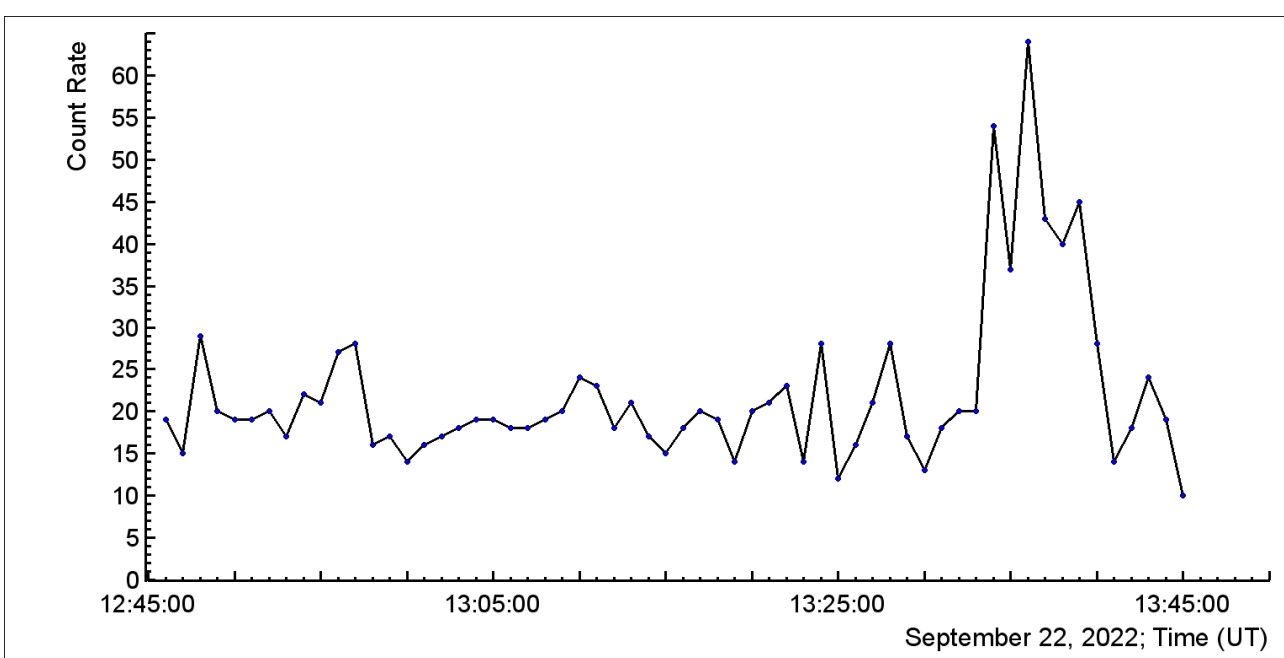

**Figure 11.** 1-min time series of count rate of coincidences of 2 identical 1 m$^2$ plastic scintillators and NaI(Tl) detector located in Nor Amberd.

## 5. Comments on the "Lightning" Origin of Enhanced Particle Fluxes from Thunderclouds

The signals from atmospheric discharges and particle detectors were monitored on Aragats to register TGE particles simultaneously with lightning flashes. The developed fast synchronized data acquisition system (FSDAQ, [40], Figure 12) was triggered by the MFJ-1022 antenna, covering a frequency range of 300 kHz to 200 MHz. A flat plate antenna with a passive integrator recorded fast electric field signals. The output of the integrator was directly connected to a digital oscilloscope (2-channel Picoscope 5244B). The data capture duration was 1 s, including 200 ms pre-trigger time and 800 ms post-trigger time. The sampling rate was 25 MS/s, corresponding to an interval of 40 ns, and the amplitude resolution was 8 bits. The trigger output of the oscilloscope was connected to the input of the GPS timing system of the myRIO board from the National Instrument [41]. Any event detected by the oscilloscope generated an output trigger, causing the GPS board to trigger simultaneously and produce a timestamp. The eight digital inputs of the myRIO board were used to feed signals from various particle detectors operating on Aragats. Signals from the electric field sensor (electric mill EFM-100) were provided to the myRIO board via TCP-IP connection (WiFi). Changes in the electrostatic field were recorded with a sampling interval of 50 ms; the amplitude resolution of electric field measurements was 0.01 kV/m, and the accuracy of lightning location was ≈1.5 km. At any trigger, the myRIO board generated a particular output signal containing the particle detector signals, the NSEF value, and the exact arrival time of the trigger signal. Thus, fast waveforms (from atmospheric discharges) were synchronized with particle fluxes and slow (20 Hz) NSEF measurements.

Time series of particle detector count rates, electrostatic field measurements, service information, and digital oscilloscope data files are transferred via an online PC to the MySQL database at the CRD headquarters in Yerevan (http://adei.crd.yerphi.am, accessed on 29 January 2023). Two independent FSDAQ systems, located in the MAKET and SKL experimental halls at Aragats (Figure 12a,b), were triggered by two independent whip antennas. During the April–June months, we recorded numerous lightning flashes and TGE events; for instance, in 2017, there were ~250 common triggers of MAKET and SKL FSDAQ systems. A careful study of the output waveforms from the flat antenna and the particle detectors showed that neither signal of a particle coincided with a lightning flash.

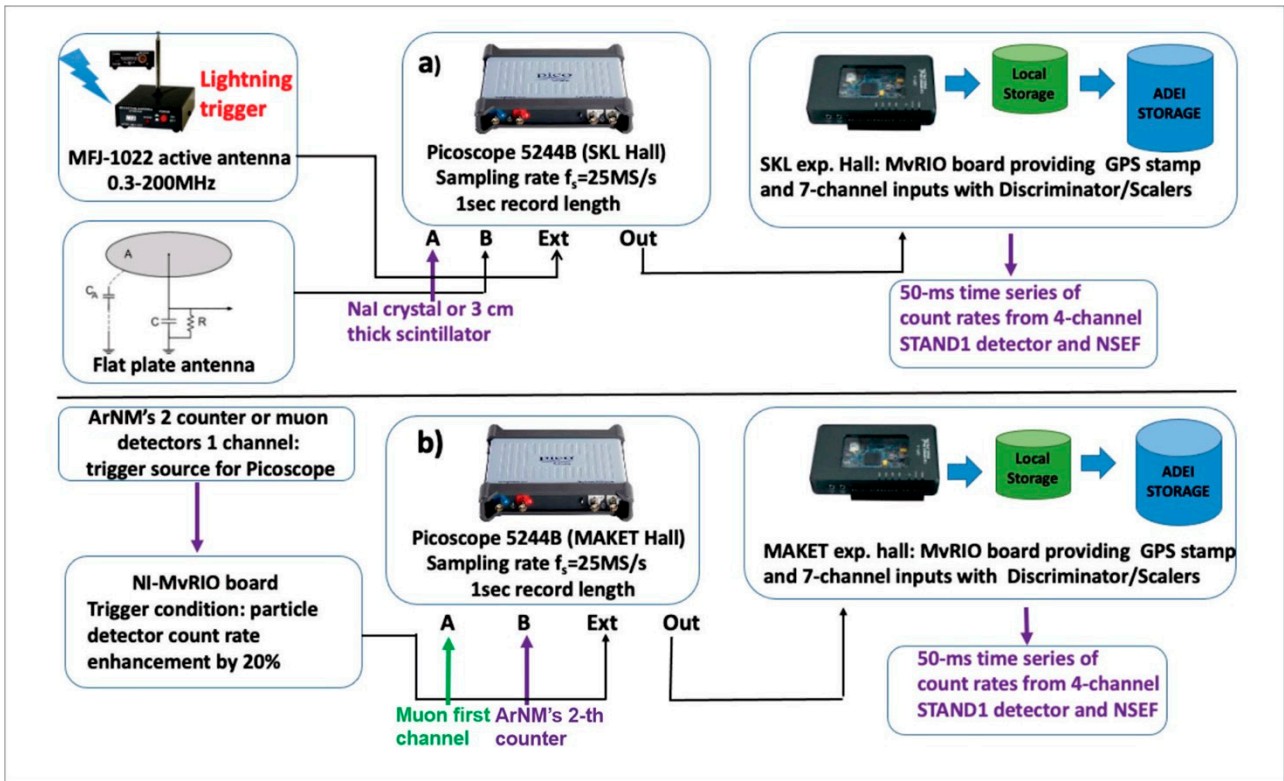

**Figure 12.** (**a**) Components of the Fast Synchronized Data Acquisition (FSDAQ) system for studying the coupling of particles and lightning flashes; (**b**) A similar system triggered by a TGE.

All registered particle "bursts" were bipolar and could be easily distinguished from the unipolar signals from particles traversing the detector. For instance, in Figure 13, a normal intracloud lightning flash abruptly terminated the TGE at 11:59:51.82.

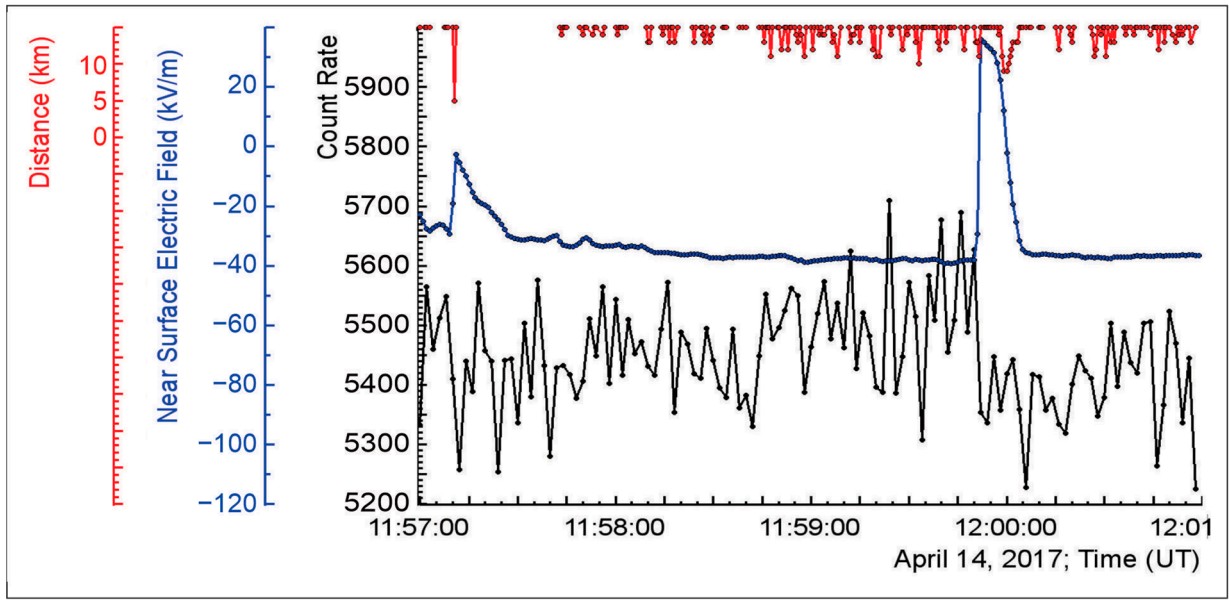

**Figure 13.** TGE abruptly interrupted by a lightning flash at 11:59:51.82; triggers were registered in the MAKET and SKL halls at 11:59:51.75; a surge of the NSEF started at 11:59:51.94; the decline of particle flux started at 11:59:51.83.

The outputs of the two plastic scintillators synchronized with the trigger are shown in Figure 14. Four bipolar signals in Figure 14a, after examining the zoomed version shown in Figure 14b, could be easily distinguished from genuine unipolar signals from particles passing through the detector. Thus, the lightning flash that abruptly terminates TGE does not initiate any particle burst; only electromagnetic interferences in the detector electronics/cabling are seen.

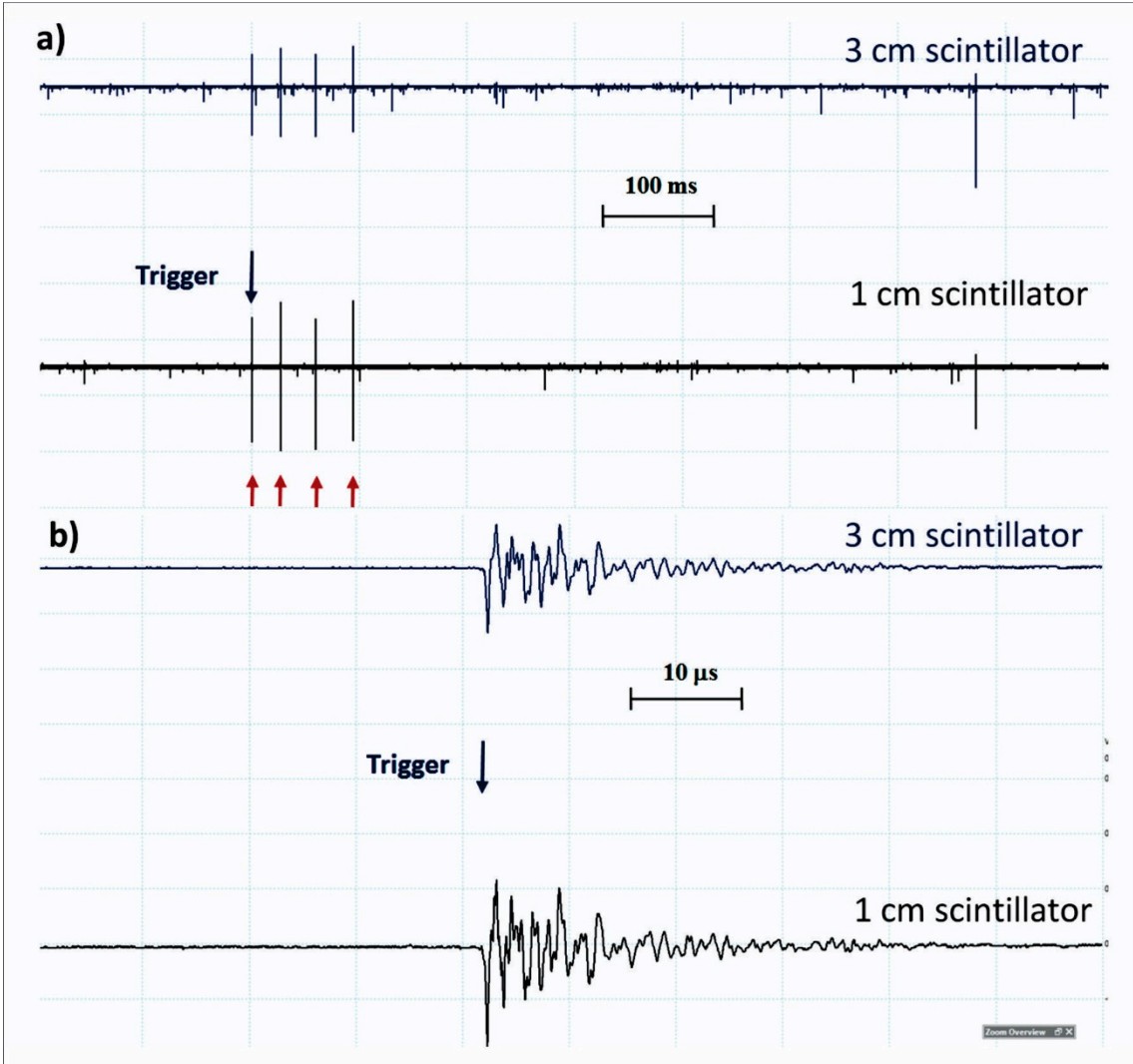

**Figure 14.** (**a**) "Particle bursts" observed on 14 April 2017 using 1 cm and 3 cm thick plastic scintillators with an area of 1 m$^2$ in the MAKET experimental hall. The red arrow indicates signals. The frame (**b**) is a zoomed-in version of the data shown in (**a**). The fast waveforms were synchronized with the atmospheric discharge at 11:59:51.75.

Thus, after many years of continuous observation of TGEs and lightning flashes, we confirm the earlier conclusion [42] that we have not found any correlation between particle fluxes with MeV energies and lightning flashes. The RREA process accelerates and multiplies electrons to energies of tens of MeV without the need of any additional mechanisms, such as the relativistic discharge mode with feedback [43].

## 6. Are EAS Cores, or Specific Lightning Discharges Generating Downward Terrestrial Gamma Flashes (DTGFs)?

A small, fast scintillation detector (7.62 × 7.62 cm LaBr3) located near the water-Cherenkov array of the high-altitude water Cherenkov facility (HAWC, [44]), consisting of 300 water Cherenkov detectors 4 m high and 7.3 m in diameter, routinely registered particle bursts. The electronics collected 15 ms of data for each trigger, with 5 ms of data before the trigger. All flashes observed at HAWC between September 2017 and September 2019 occurred on days with fair weather, meaning there were no lightning flashes nearby; see Table 1 from [45]. CORSIKA simulations show that bursts are generated by EAS core particles captured in nuclei of soil which produce high-energy gamma-rays through (n, γ) reactions.

Another large particle detector, the telescope array's (TA's) surface detector (TASD, [46]), also registered particle bursts, which authors of [47] relate to the special lightning flashes. TA consists of 507 units on a 1.2 km square grid covering a total area of 700 km$^2$. Each unit comprises upper and lower scintillators 1 cm thick and 3 m$^2$ in size. TASD triggers are recorded at 1-millisecond time intervals in correlation with lightning flashes registered by the lightning mapping arrays (LMA) and Vaisala single lightning detector network (NLDN). Typically, NLDN records about 750 flashes (intracloud and cloud-to-ground) per year above 700 km$^2$ of TASD. Over 8 years of operation, 20 particle bursts associated with thunderstorm activity were detected [47]. Thus, less than 0.7% of flashes reported over TASD were accompanied by identifiable particle bursts. The authors of [48] attribute these bursts not to EAS, as in the HAWC and Aragats experiments, but to the downward negative leaders, which end up in a negative cloud-to-ground discharge. In contrast, particle bursts and thunderstorm activity are separated in the HAWC and Aragats experiments.

The difficulty of detecting EAS cores results from their propagation through the detector taking only a very short time (a few tens of ns). Usually, the window for particle registration is set to ≈1 microsecond, and all EAS core particles will be registered as one huge pulse. Yuri Stenkin et al. [49] described the indirect detection of the EAS core particles using a neutron monitor. High-energy hadrons and gamma rays from the EAS core generate numerous thermal neutrons in the soil and detector material, increasing the NM count rate (neutron multiplicity). This possibility of the EAS core detection by NM was almost not recognized in the past because the commonly used long dead time does not permit counting the neutron multiplicity. By setting ≈3000 times shorter dead time of 0.4 μs at Aragats neutron monitor (ArNM), we counted all thermal neutrons entering ArNM. In this way, ArNM registered large multiplicities exceeding 2000 (see Figures 20–22 of [50]). The ArNM signal was sent to the FSDAQ (see Figure 12b) to trigger the oscilloscope when the detector's count rate exceeded a specified threshold (typically an increase of 20% above the moving average). In Figure 15a–d, we show pulses of a neutron burst with a multiplicity of 107, recorded at 4:08:05 on 26 November 2016. The bursts were observed as a sequence of microsecond pulses isolated in time from other pulses by at least 100 microseconds.

Thus, the neutron monitor increases the EAS core time profile (20–30 ns) by approximately 5 orders of magnitude (2–3 milliseconds), which makes it possible to use a relatively slow device (the NM) to detect particle bursts. Therefore, "downward TGFs" can be explained by a widespread phenomenon in cosmic ray physics, namely by EAS core hitting the detector. Possibly, the bursts registered by TASD, are also initiated by the EAS cores without any connection to thunderstorm activity. Comprehensive information about the EAS cores hitting ArNM can be found in the dataset of 50 high multiplicity events published in the Mendeley repository [51].

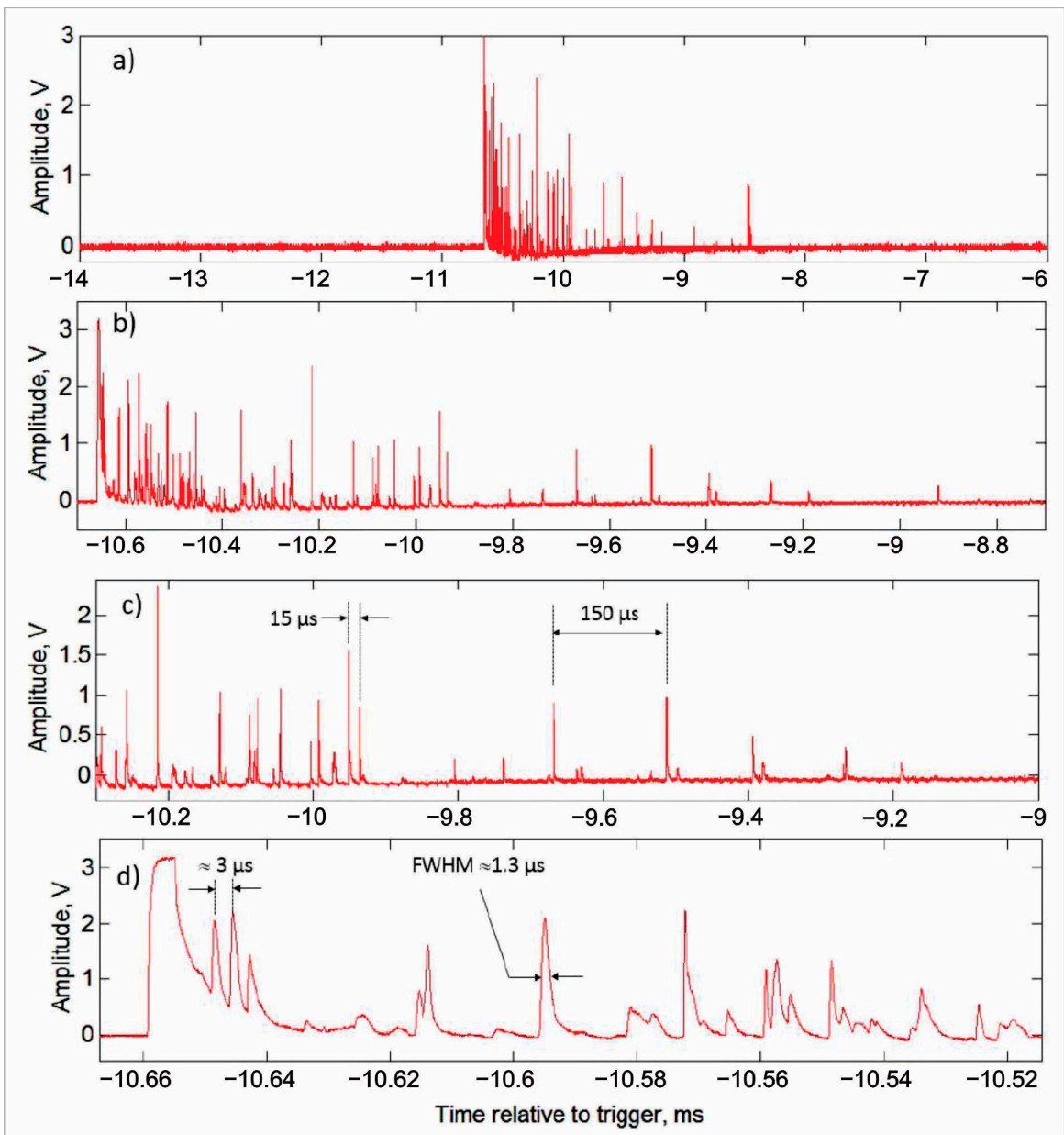

**Figure 15.** Oscillograms of the neutron burst that occurred at 04:08:05 on 26 November 2016. Flash duration of 2.2 ms and multiplicity of 107. Four panels (**a–d**) show burst records on different time scales. ©New Astronomy. Reproduced with permission.

## 7. Overestimation of the Gamma Ray Energy by High-Altitude EAS Arrays during Thunderstorms

With the creation of high-altitude EAS facilities HAWC [44] and LHAASO [52] with excellent capabilities for separating gamma rays from the hadronic background, the century-old problem of the origin of cosmic rays is finally close to being solved. For the first time, verifying that the galactic gamma radiation spectra measured from the direction of several known super-novae remnants extend beyond 1 PeV was possible. Due to its 1 km$^2$ detector surface, including large muon facilities, LHASSO has a low energy threshold ($\approx$1 TeV) and excellent suppression of extensive hadron-induced air showers. We chose the LHASSO array for our analysis because they recently identified 12 PeVatron candidates from celestial sources previously observed with atmospheric Cherenkov telescopes. The LHAASO site is located on Mount Haizi, Daocheng County, Sichuan Province, on the Tibetan Plateau's edge, at an altitude of 4410 m above sea level. The Tibetan Plateau is known to be a place

of frequent thunderstorms and huge intracloud electric fields, the vertical profile of which can reach 1–2 km, with a strength of 1.5–2 kV/cm.

Several EAS experiments, including those in Tibet, have already reported a 20–30% increase in triggering frequency during thunderstorms [9–11]. Therefore, RREAs can effectively mimic EAS by successfully passing all the software checks. At Aragats, on 19 September 2019, a 400% increase in the frequency of EAS triggers was registered [2,53]. To understand the influence of a strong electric field on the measured EAS size (number of electrons), we performed a simulation study using the CORSIKA code [36]. The electric field was introduced at altitudes of 4460–6460 m, and secondary particles (hadrons, muons, electrons, and gamma rays) were tracked until their energies decreased to 0.3, 0.3, 0.03, and 0.03 GeV, respectively. Table 2 shows the number of electrons in fair weather and how EAS electrons increased dramatically after crossing the atmospheric electric field.

**Table 2.** Enhancement of the number of electrons initiated by a primary gamma ray with energies from 1 to 1000 TeV after passing the electric field of different strengths.

| Eo (TeV) | Ne | | | |
|---|---|---|---|---|
| | Ez = 0 kV/cm | Ez = 1.9 kV/cm | Ez = 2.0 kV/cm | Ez = 2.1 kV/cm |
| 1 | 316 | 12,103 | 15,904 | 18,044 |
| 10 | 5560 | 148,088 | 201,096 | 229,163 |
| 100 | 69,996 | 1,374,853 | 1,775,837 | 2,169,369 |
| 1000 | 827,547 | 10,346,388 | 13,605,357 | 14,066,929 |

Figure 16 shows a sharp increase in the number of electrons after increasing the electric field strength above the critical value. Starting from 1.7 keV/cm, the number of electrons grows exponentially for all energies of primary gamma rays.

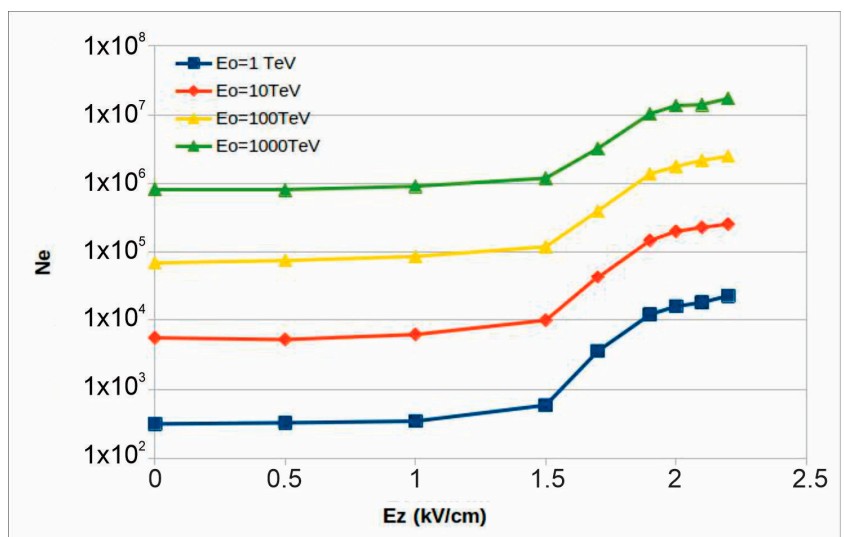

**Figure 16.** The number of electrons registered on the Earth's surface after crossing the atmospheric electric field of different strengths. The primary gamma ray enters the electric field at an altitude of 6460 m.

As the number of highest energy gamma rays observed by LHAASO is relatively low, it is essential to check if they were detected in fair weather, if electrons were multiplied in the strong electric fields above the detector, and if energy estimation is positively biased. In Table 3, we show the energy of a primary gamma ray used in the simulation and calculated using the "measured" number of electrons after EAS crosses the atmospheric electric field (as an energy estimator, a simple linear function was used that connected the shower size and the primary gamma ray energy). As can be seen from the Table 3, the calculated energy

of the primary gamma rays differs significantly from the "true" values. Thus, for the low primary energies (1 TeV), the bias in primary gamma ray energy estimation can be ten-fold and more, and for the higher energies (1 PeV), it can be ≈2.5 times.

**Table 3.** "True" and estimated energies of primary gamma rays after passing through the electric field of 2.1 kV/cm strength.

| Eo (GeV) | Eest (GeV) |
|---|---|
| $10^3$ | $2.23 \times 10^4$ |
| $10^4$ | $1.34 \times 10^5$ |
| $10^5$ | $6.50 \times 10^5$ |
| $10^6$ | $2.42 \times 10^6$ |

## 8. Forbush Decrease Measured by SEVAN Network

The solar wind regularly modulates the flux of low-energy galactic cosmic rays, which changes the structure and polarity of local magnetic fields in the heliosphere and magnetosphere. Thus, variations in the intensity of secondary cosmic rays observed on the Earth's surface can provide valuable information on the distribution of these structures in the heliosphere and on the interaction of the solar wind with the magnetosphere. Short-term variations are observed in the GCR flux, caused by the passage of the "fast" solar wind, shock waves, and magnetized solar plasma ejected from the Sun (interplanetary coronal mass ejection (ICME)). Particle fluxes measured on the Earth's surface show depletion (called Forbush decrease (FD)) and enhancement (geomagnetic effects) due to disturbances in near-Earth magnetic structures in response to propagating shocks and ICME. Forbush decreases are the most frequent and easily detectable phenomenon of the solar modulation of galactic cosmic rays. Historically more than eighty years ago, Scott Forbush was the first to link these CR depletions to solar eruptions [54].

When observed with particle detector networks, FDs show an asymmetric structure: a rapid decrease in flux within several hours, followed by a gradual recovery with a time scale of several days. The cosmic ray flux usually shows a preliminary increase of about 1 to 2% due to the reflection of cosmic rays from the approaching solar wind shock. If solar flares follow one after another (usually from the same active region), then several fast waves of magnetized solar plasma propagate in the interplanetary space simultaneously and sometimes catch up with each other, so the FDs can have a rather complex structure, demonstrating sequential depletion without a recovery stage.

On 3–5 November 2021, the largest FD of the current solar activity cycle produced geomagnetic storms and auroras as low as New Mexico, USA (+34N). SOHO coronagraphs recorded a plasma cloud leaving the Sun on 2 November, following and overtaking a slow solar flare (M1.7) in the magnetic canopy of sunspot AR2891. SEVAN network [55] registered this FD at the nodes in Aragats, Lomnicky Shtit, Musala, and Hamburg [56]; see Figure 17. Particle flux depletion was approximately 10%. FD at mountain heights is better expressed than at sea level (Hamburg).

Figure 18 shows measurements of disturbances in the electric and geomagnetic fields observed when the ICME reaches our planet. We show that the X-component of the geomagnetic field reaches its minimum value around 8:00 UT (blue curve). The NSEF disturbances of −500 V (black curve) correlate well (correlation coefficient ≈0.6) with the geomagnetic field disturbances, with a delay of ≈50 min; see the inset.

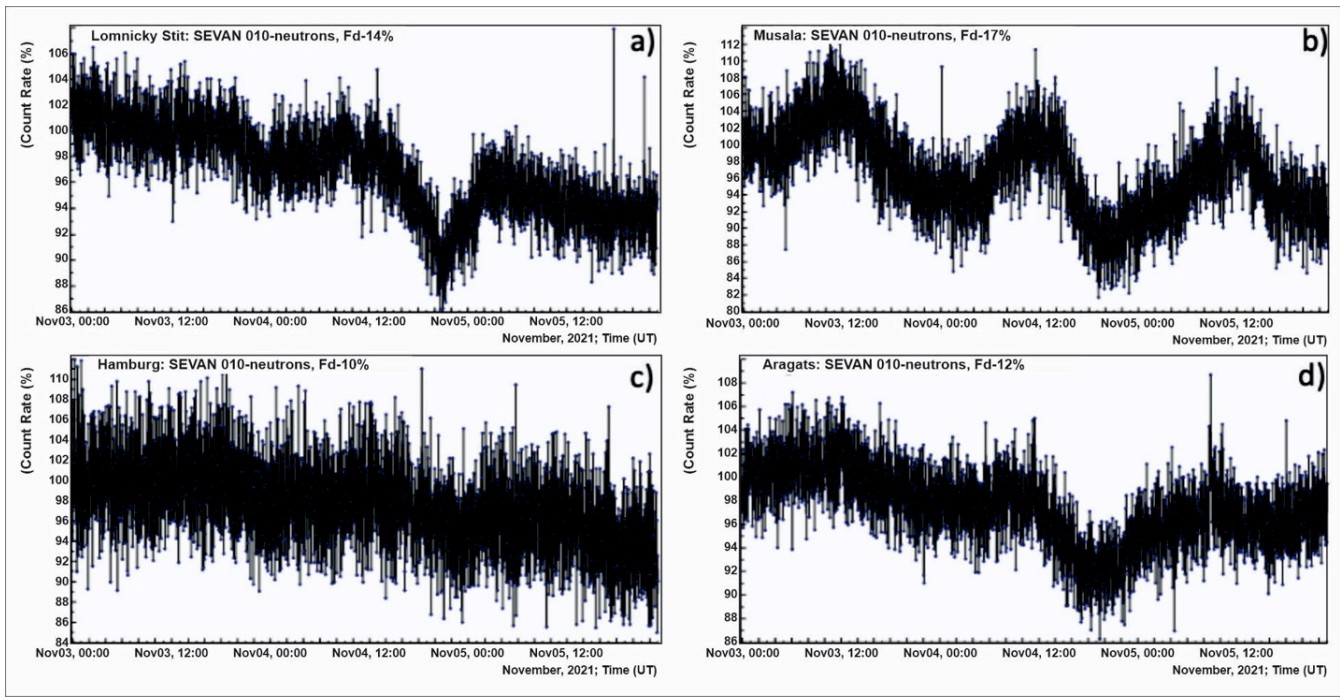

**Figure 17.** 1 min time series of count rates of "010" coincidence of SEVAN layers (primarily neutrons) for 4 nodes in Slovakia (**a**), Bulgaria (**b**), Germany (**c**), and Armenia (**d**).

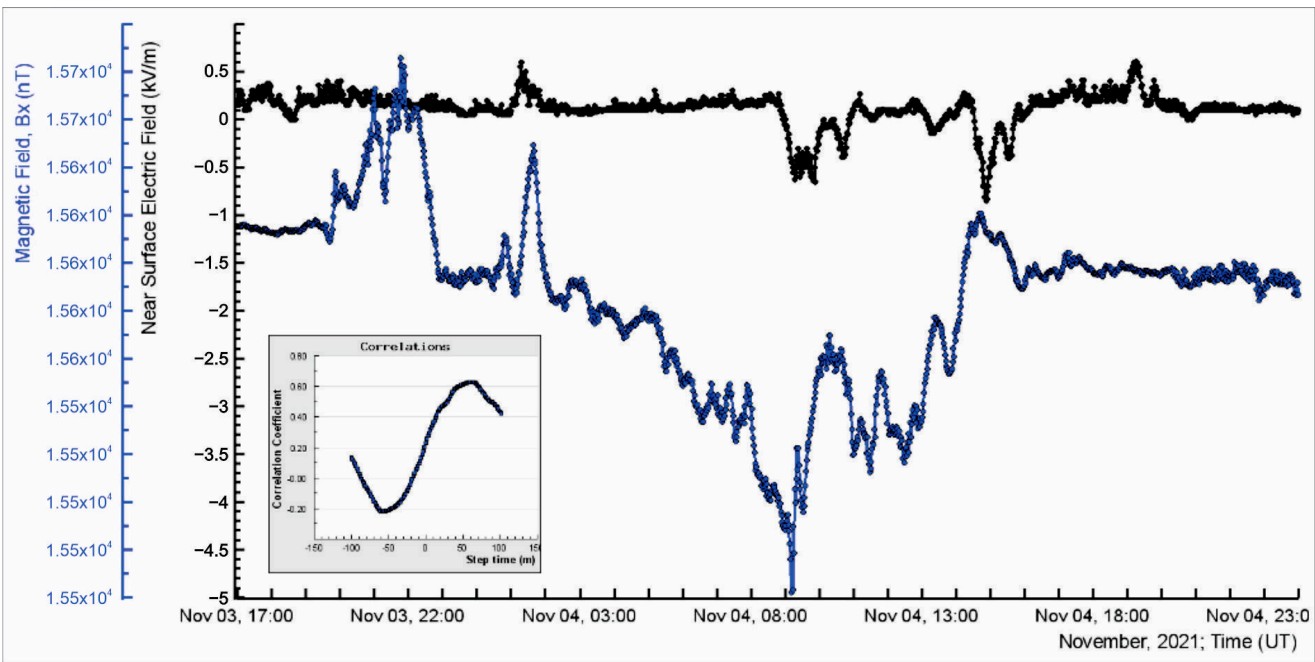

**Figure 18.** Disturbances in the X-component of the geomagnetic field (blue) and NSEF (black) during the FD. The inset shows a curve of lagged correlations; NSEF disturbances lag relative to geomagnetic field disturbances by ≈50 min.

## 9. Estimation of the Maximum Strength of the Atmospheric Electric Field on Lomnicky Stit

Understanding the maximum potential difference (voltage) inside a thunderstorm is one of the fundamental problems of atmospheric physics, directly related to the mystery of lightning and electron acceleration [57]. Numerous measurements on balloons, airplanes, and mountain peaks confirm that a lightning flash usually abruptly terminates particle

fluxes. The identical SEVAN particle detectors that observed the Forbush decrease on 4 November 2021 regularly measured TGE in the highest mountains of Eastern Europe. The electric field measured on sharp peaks is stronger than on the Aragats highland. More than 150 TGEs observed at Aragats were terminated by a lightning flash when the absolute value of the NSEF was sufficiently high [58,59]. Thus, the energy spectra of TGE electrons and gamma rays, measured at the TGE flux maximum before the lightning, contain information about the strength of the electric field that initiates the TGE and then the lightning flash. We performed a cycle of simulations of the RREA process in the atmosphere to test the conditions leading to the maximum particle fluxes and maximum particle energies in the RREA, which are directly related to the maximum achievable voltage in the thunderous atmosphere.

The maximum voltage measured at Aragats was ≈300 MV [60] at the steep top of the Lomnitsky Shtit; the maximum voltage was estimated from the huge TGE of 10 June 2017 [61]. The count rates of this TGE are shown in Figure 19 and Table 4. Table 4 shows the average of the 1 min count rates measured before the event and the count rates measured at maximum flux. The count rates of the SEVAN's upper scintillator and the coincidence "100" exceed the count rates of fair weather more than a hundred times (last column of Table 4).

**Table 4.** Minute average count rates of SEVAN and NM detectors located at Lomnicky Stit and extreme values at the maximum flux minute of a TGE registered on 10 June 2017.

| Name | Mean 1/min | σ | 13:14 1/min | % | N Times |
|---|---|---|---|---|---|
| SEVAN upper | 25,047 | 171 | 2,534,000 | 10,013 | 101 |
| Coincidence "111" muons | 1929 | 48 | 1666 | −14 | |
| Coincidence "100 "low energy part. | 19,550 | 142 | 2,526,000 | 12,890 | 130 |
| Coincidence "010" gamma rays | 1468 | 39 | 3326 | 125 | 2.7 |
| Neutron monitor | 29,640 | 265 | 71,220 | 140 | 2.4 |

This world's largest TGE reaches its maximum in one minute. After that, an intense flow of electrons initiates a lightning flash, which stops the TGE. The TGE ended with a complicated multicycle discharge registered by the EUCLID network at 13:14:35 [62], at a distance of 1 km from the detector. The depletion of the muon flux (muon stopping effect, [60]) in the same minute was ≈14% (see Figure 19b), which is twice that observed during the strongest event at Aragats on 4 October 2010 [2]. The oblique muon depletion was much larger (45%); see Figure 19c.

The enhancements of the SEVAN "010" coincidence and NM count rates do not reach these extreme values but are much greater than those measured at Aragats [63]. Two independent detectors measured a vast increase in the neutron flux in the same place and at the same time. The data from the NM (140% enhancement) are essential as evidence of photonuclear reactions of high-energy gamma rays produced in the TGE (the previous maximum flux recorded by the neutron monitor at Aragats was only 5.5% [1]). This result unambiguously proves the photonuclear origin of atmospheric neutrons.

Proceeding from the TGE parameters shown in Figure 19 and Table 4, we performed CORSIKA simulations to estimate the atmospheric electric field that can enable such a vast RREA, which reaches the Earth's surface and generates enormous TGE. This time, we calculate the detector response for the direct comparison of simulations with the count rate measured by the upper scintillator of the SEVAN detector.

The simulation of the RREA was performed for strengths of the atmospheric electric field of 2.3–2.5 kV/cm within heights of 2.6–4.6 km, where the uniform electric field was introduced with strength exceeding the critical value by 10–40% (Figure 20). Uniformity of the electric field extending 2 km leads to changing the actual surplus of the critical energy at different heights according to the air density value. Thus, the 2.4 kV at the altitude of 4.6 km is ≈32% larger than the critical energy, whereas, at the height of 2.6 km, it is only ≈15% larger. In Figure 20, the number of electrons and gamma rays was integrated from

7 MeV to be compared with the SEVAN upper scintillator's count rate, for which the energy threshold is ≈7 MeV.

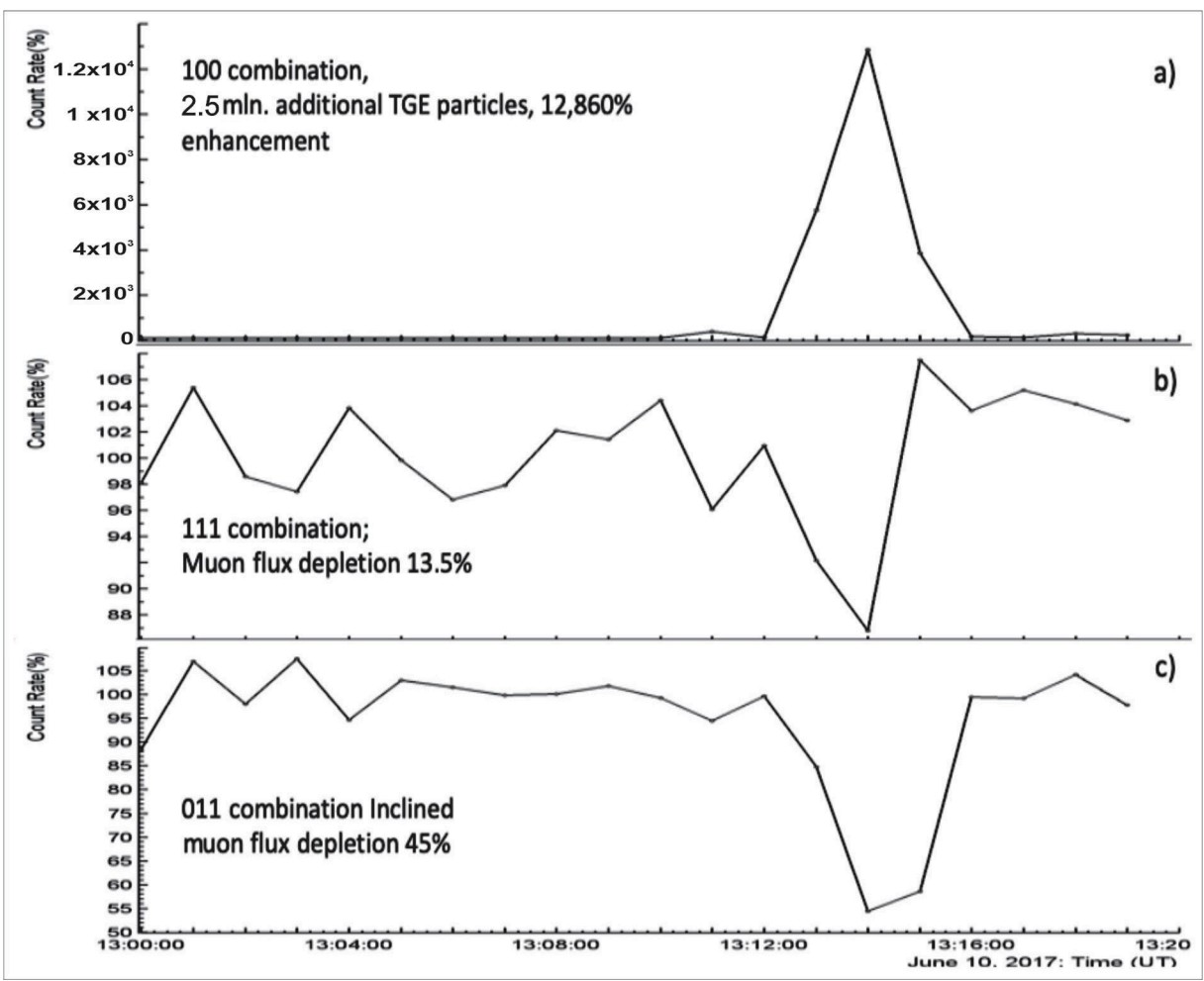

**Figure 19.** Extreme TGE event detected by SEVAN detector located on Lomnicky Stit: (**a**) TGE particles—electrons and gamma rays; (**b**) high energy muons; (**c**) inclined muons.

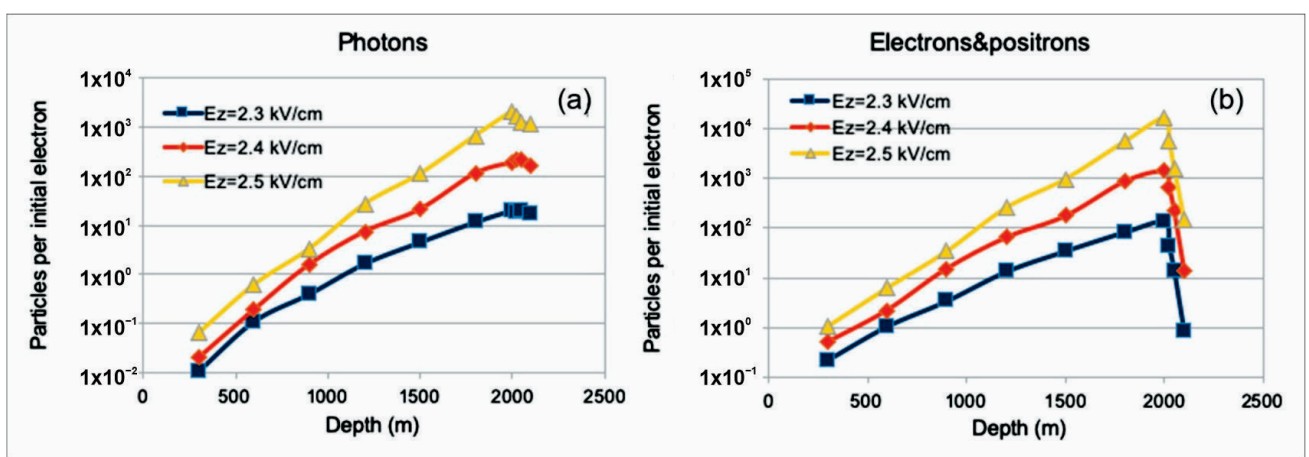

**Figure 20.** Development of RREAs in the thunderous atmosphere. The avalanche started at an altitude of 4600 m, 2 km above the SEVAN detector at Lomnicky Stit. The number of avalanche particles was calculated every 300 m per seed electron per $m^2$. After leaving the electric field, the avalanche particles were tracked for another 100 m.

In Table 5, we show the expected count rates of the upper scintillator of the SEVAN detector corresponding to the avalanche particle numbers shown in Figure 20. The numbers of electrons and gamma rays are multiplied by the efficiency of electron and gamma ray registration in a 5 cm thick plastic scintillator, which is ≈95% for electrons and positrons and ≈5% for gamma rays. The first three columns of Table 5 show the simulation results for four (many more were conducted) configurations of the electric field in the thundercloud, namely, the number of electrons and gamma rays registered by SEVAN upper scintillator per seed electron. In the fourth column, we show the number of RREA particles expected to be registered by this scintillator per second per seed electron. The last column shows the total number of model particles registered by the SEVAN detector per second (multiplied by the number of seed electrons). The number of seed electrons on 4600 m was calculated using the EXPACS calculator [38] and equaled 455 per m$^2$ per second. The last row of Table 5 shows the number of TGE particles measured by the SEVAN upper scintillator at 13:14 UT per second (see the first row of Table 4, where the count rates are given per minute). We show the count rates in Table 5 per second because per-minute count rates are too large.

**Table 5.** Expected SEVAN upper detector count rates (modeled counts multiplied by registration efficiencies, per seed electron, per second) at Lomnicky Stit for different electric fields, and the measured count rate of the upper SEVAN scintillator at 13:14, 10 June 2017 (last row). In the previous column, simulated counts are multiplied by the number of seed electrons.

| SEVAN Upper Scintillator | Electron Counts /m$^2$ s | Gamma Ray Counts /m$^2$ s | Sum El. + Gamma /m$^2$ s | Total Expected Counts /m$^2$ s |
|---|---|---|---|---|
| 2.4 kV/cm 50 m | 175 | 13 | 188 | 85,540 |
| 2.4 kV/cm 100 m | 11 | 10 | 21 | 9555 |
| 2.5 kV/cm 50 m | 1268 | 76 | 1344 | 611,520 |
| 2.5 kV/cm 100 m | 119 | 68 | 187 | 85,085 |
| TGE on 10 June 2017 | | | | 42,223 |

The estimated count rate of 85,540 was obtained for the field strength of 2.4 kV/cm if the electric field ended at 50 m above the Earth's surface. For the field strength of 2.5 kV/cm, we obtained from simulations 85,085 particles reaching the Earth's surface if the electric field ended at 100 m. If we assume that the electric field is prolonged to 50 m, we obtain ≈ten times more particles, and this electric field cannot be reached because, due to an enormous electron flux, a lightning flash will abruptly terminate the RREA [59,64]. Therefore, an electric field cannot be larger than 2.5 kV/cm. Thus, we estimate the maximum electric field strength to be 2.5 kV/cm and voltage to be 2.5 kV/cm × 2 km = 500 MV.

We recognize that the relation between the electric-field strength and TGE particle fluxes is nonlinear and depends on many unknown parameters of the atmospheric electric field and meteorological conditions. However, substantial particle fluxes measured by the SEVAN detector allow us to obtain a reasonable estimate of the maximum electric field, choosing the appropriate field strength and its spatial extent from several alternatives obtained in the simulation trials.

## 10. Conclusions

TGE measurements (see the catalog of ≈600 TGEs in [65])prove that a strong electric field covers enormous volumes in a thunderstorm atmosphere. The largest TGEs registered by the SEVAN network [55] on Mount Musala (Bulgaria) and Mount Lomnitsky Shtit (Slovakia [61]), as well as the results obtained by the Japanese group [28], prove that TGE is not a specific feature of Aragats mountain, but is a universal characteristic of thunderstorms. The measured energy spectra unambiguously connect the RREAs developed in the

thunderous atmosphere and TGEs registered on the Earth's surface. Monitoring particle fluxes makes it possible to estimate the maximum potential drop (voltage), reveal the charge structure of a thundercloud, and clarify the role of the LPCR in developing TGE.

RREAs are accelerated to high energies by atmospheric electric fields and turn into TGEs, which release significant doses of radiation to the Earth's surface. This additional radiation must be introduced into weather forecasting and global change models. Monitoring the electrons and gamma ray fluxes shows a large extent of strong electric field vertically and horizontally. Multiple surface detectors can simultaneously monitor atmospheric electric fields. The temporal resolution is on the order of seconds, and monitoring is carried out around the clock and seven days a week without the possibility of missing strong storms. Especially important is that monitoring can disclose powerful electric fields, which can reach high values just tens of meters above the Earth's surface. Atmospheric remote sensing can be used with field mill climatology, which typically saturates in strong electric fields, to minimize the risk of launching spacecraft during thunderstorms [66].

Recently, large particle arrays have paid more and more attention to atmospheric conditions and emerging electrostatic fields. At the Pierre Auger Observatory, a network of electric mills was installed to research the radio emission of air showers in strong atmospheric electric fields during thunderstorm conditions [67]. The Major Atmospheric Gamma-ray Imaging Cherenkov (MAGIC) telescopes make calorimetric use of the Earth's atmosphere for measuring the energy of the very-high-energy gamma quanta. Thus, the accuracy of energy estimates strongly depends on the quality of the atmosphere at the time of the observations. For introducing corrections for atmospheric conditions, a single wavelength elastic LIDAR (LIght Detection and Ranging) system performs real-time ranged-resolved measurements of aerosol transmission [68]. The IceCube detector samples trillions of muon events at extreme temperature variations in the stratosphere. The recovery of the atmospheric profile for the production of muons led to a better understanding of seasonal variations of neutrino flux. The muon flux measurements also reveal the nonlinear relationship between the muon rate and the effective temperature [69].

Our measurements show that a strong electric field can highly amplify the cosmic ray flux over large areas of particle detector arrays, significantly changing the number of EAS electrons and consequently leading to overestimating the energy of primary particles.

The TGEs and short particle bursts observed on the Earth's surface have roots in EAS physics, the first using free electrons as seeds for RREA and the second emerging from the EAS core hitting the detector area. The particle bursts observed in the HAWC and Aragats experiments can be explained using traditional EAS physics.

Cosmic ray physics and HEPA are synergistically related, and results need to be exchanged to explain particle bursts and reveal the influence of atmospheric electric fields on the shape and size of EASs. Particle fluxes measured by spatially scattered networks, combined with information from satellites, provide experimental data on the most energetic processes on the Sun and in the Earth's atmosphere and will become an essential element of a global space weather monitoring and forecasting service. Thus, precise monitoring of the particle fluxes on the Earth's surface with consequent recovering fluxes and energy spectra leads to a greater understanding of atmospheric modulation effects and, consequently, to greater knowledge of the structure and strength of the atmospheric electric fields.

**Funding:** This research was funded by the Science Committee of the Republic of Armenia for their support (scientific project No. 21AG-1C012).

**Institutional Review Board Statement:** Not applicable.

**Informed Consent Statement:** Not applicable.

**Data Availability Statement:** The data for this study are available in numerical and graphical formats using the ADEI multivariate visualization platform on the web page http://adei.crd.yerphi.am/adei, (accessed on 29 January 2023).

**Acknowledgments:** This review is based on recent studies published by the physicists of the Cosmic Ray Division of Yerevan Physics Institute, to which the author expresses his deep gratitude. He thanks the Aragats Space Environmental Center team for the uninterrupted operation of the experimental facilities at Aragats in severe weather conditions. The author expresses his gratitude to the Science Committee of the Republic of Armenia for their support (scientific project No. 21AG-1C012) in the modernization of the technical infrastructure of high-altitude stations.

**Conflicts of Interest:** The authors declare no conflict of interest.

## Nomenclature

| | |
|---|---|
| GCR | Galactic cosmic rays |
| HEPA | High-energy physics in the atmosphere |
| RREA | Relativistic runaway electron avalanche |
| TGE | Thunderstorm ground enhancement |
| NSEF | Near-surface electrical field |
| NGR | Natural gamma radiation |
| MN-MIRR | Dipole made by the main negative layer and its mirror in the Earth |
| MN-LPCR | Dipole made by the main negative and lower positively charged layers |
| TGF | Terrestrial gamma flash |
| DTGF | Downward TGF |
| EXPACS | EXcel-based Program for calculating Atmospheric Cosmic-ray Spectrum |
| CORSIKA | COsmic Ray SImulations for KAscade |
| BOLTEK | Company producing EFM-100 electric field sensor |
| ASNT | Aragats Solar Neutron Telescope |
| STAND1 | Particle detector network on Aragats |
| HAWK | High-altitude water Cherenkov facility |
| TA | Telescope Array |
| TASD | Telescope array's scintillator detector |
| WWLLN | Worldwide lightning location networks |
| NLDN | Vaisala single lightning detector network |
| LMA | Lightning mapping array |
| FSDAQ | Fast synchronized data acquisition system |
| CME | Coronal mass ejection |
| ICME | Interplanetary CME |
| GMS | Geomagnetic storm |
| FD | Forbush decrease |
| FWMH | Full width on half maximum |
| MAKET | Experimental hall on Aragats |
| SKL | Experimental hall on Aragats |

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
