# Peer review of "Thunderstorm Ground Enhancements Measured on Aragats and Progress of High-Energy Physics in the Atmosphere"

_atmosphere, doi:10.3390/atmos14020300_

Round 1

Reviewer 2 Report

The article presents an important summary of the study of the atmospheric electric field, especially during thunderstorms. The work is original and should be published. The following is a list of recommended revisions:

1. There are TOO many acronyms, especially for a review article. The reviewer had to make a list of acronym definitions in order to follow the context. The most important acronym is TGE, which is actually not introduced. Please make an effort to reduce the use of these so that general readers can follow with less frustration.

2. Use consistent notation. For example, either use Fig. X or Figure X. In general, it is recommended to use Figure X at the beginning of a sentence and use Fig. X in the middle of a sentence. The same for Table. X vs Table X.

Also, use the consistent font for "Figure" in figure captions. In addition, be consistent with 1 s or 1 second, 1-second, one second, and in the similar case for ms vs millisecond.

3. The first column of Table 1 is confusing, i.e., mixing field strength with dates.

4. A collection of minor corrections should be made:

Line 33, change "." to "," before coinciding

Line 130, define MN. Why simply spell it out.

Line 179, what is RGEA?

Line 204,  "RREA" should be "ASNT"

Line 225, "pr second" -> "per second"

Line 250, "2022" -> "2009"

Line 409, "TGFs" ?

Line 411, "from" -> "of"

Line 429, remove "." after [48]

Line 438, the paragraph is too short

Line 412, remove "m."

5. Most of the plots style are different from each other.

Reviewer 3 Report

The article provides a comprehensive overview of the results obtained from long-term observations of cosmic radiation in correlation with the state of the atmosphere at Aragats station. The subject is relatively new and studies such as this are therefore very valuable. 

It should definitely be presented to a wide audience 

Reviewer 4 Report

The paper “The synergy between High-energy Physics in the Atmosphere and Cosmic Ray Physics” by Ashot Chilingarian is the review where data and results obtained at arrays and observatories in Armenia are presented along with data of arrays worldwide.

The paper can be accepted after the revision of abbreviations.

There is a lot of them in the text, but not each is decoded. I recommend to include the list of abbreviations in the text, possibly before Introduction.

The list of remarks is in attachment..

Reviewer 5 Report

According to the title and the abstract, the paper is supposed to be a review of the relations between cosmic rays and atmospheric physics. However, the paper has a much narrower focus on TGEs and the relation of cosmic rays to thunderstorms. As such, it is primarily a review of the author’s own results. Keeping that narrower focus in mind, the review is well written and scientific sound.

Nonetheless, prior to publication there must be a change in either title+abstract, to reflect the narrower scope, and/or an extension of the paper to really broadly review relations between atmospheric and cosmic-ray physics.

These relations between atmospheric and cosmic-ray physics extend much beyond TGEs. Just to give a few examples:

-          Seasonal variations of muons in IceCube: https://arxiv.org/abs/1909.01406

-          Enhanced radio emission of EAS during thunderstorms at LOPES: https://arxiv.org/abs/1303.7068

-          LOFAR using that EAS radio emission to investigate electric fields in thunderclouds: https://arxiv.org/abs/1703.06008

-          Study of ELVES at the Pierre Auger Observatory: https://inspirehep.net/literature/1788416

-          Relation of atmospheric aerosols and cosmic-ray measurements at the Pierre Auger Observatory: https://arxiv.org/abs/1405.7551

-          Various impacts on how atmospheric conditions impact EAS observations (many publications), and how this is mitigated, e.g., by using the GDAS atmosphere https://arxiv.org/abs/1201.2276

There may be many more examples I am not aware of. A review article on synergies between HEPA and Cosmic Rays should at least briefly mention all these topics to cover the breadth of the field.

A few further observations to be revised:

-          ‘dangerous’ processes is undefined in the abstract

-          It would be helpful to have a table of abbreviations since review articles are often read by newcomers to the field who do not yet know these abbreviations.

-          Almost all figures: Please clarify whether this is genuine research published here for the first time or whether these figures are taken from another article. In the latter case, which is totally appropriate for a review, a reference should be included in each figure caption.

-          Line 60: The statement needs to be corrected. These are neither the ‘largest’ cosmic-ray arrays, which would be first the Pierre Auger Observatory and second the Telescope Array, nor are these all operating. EAS-TOP stopped operation more than a decade ago.

In summary, I think the content is worth publication, but after a) the scope is extended as appropriate for a review, and b) the title and abstract reflect the particular focus on TGEs and thunderstorms.

Round 2

Reviewer 1 Report

Please see the attached file below that contains both my review of the revised text, and my detailed replies to the authors initial responses to suggestions.

Reviewer 5 Report

Thanks for your willingness to change the abstract. However, the change still falls short of the main point.

Title and abstract still suggest that are general review of connections between cosmic-ray detection and atmospheric physics is provided. Of course, there is no need to include all cosmic-ray physics done at IceCube, Auger, etc. in such a review. However, there are specific studies related to atmospheric physics at these and other cosmic-ray observatories.

All examples listed in my previous report are directly to about studies related of atmospheric physics, e.g., the seasonal variations in IceCube reflect seasonal variations in the atmospheric conditions, and the radio measurements of cosmic-ray air showers at LOFAR have been used as a probe of atmospheric electric fields during thunderstorms - a topic which is extensively discussed in the manuscript.

I don’t think it is appropriate in a review to not even mention the contributions to atmospheric science by several of the major cosmic-ray detectors in the world. A general review of the field should at least recognize that connections between atmospheric and cosmic-ray physics are a currently studied in several other experiments in the world.

The alternative would be say in the title and abstract that this is a review of the results at Aragats, instead of suggesting that it is a review of the complete field. Nobody doubts that the Aragats measurements are one of the most important contributions to the field. Nonetheless, following scientific standards, all important contributions to the field should be cited in the review. I don’t see any justification why these studies related to atmospheric physics at other cosmic-ray detectors are completely omitted.

Author Response

Agree; the title of the paper and abstract are changed to emphasize the topic of the review directly connected with measurements of the fluxes of the secondary cosmic rays for using them as probes of the charged structures emerging in the thunderclouds and other issues of atmospheric physics.

The atmospheric-related research performed by large cosmic ray arrays is described, and references are added.

thanks for the comments!